# RETRIEVAL INFORMATION INJECTION FOR ENHANCED MEDICAL REPORT GENERATION

## ABSTRACT

Automatically generating medical reports is an effective solution to the diagnostic bottleneck caused by physician shortage. Existing methods have demonstrated exemplary performance in generating high-textual-quality reports. Due to the high similarity among medical images as well as the structural and content homogeneity of medical reports, these methods often make it difficult to fully capture the semantic information in medical images. To address this issue, we propose a training-free Retrieval Information injectioN (RIN) method by simulating the process of Multidisciplinary Consultation. The essence of this method lies in fully utilizing similar reports of target images to enhance the performance of pre-trained medical report generation models. Specifically, we first retrieve images most similar to the target image from a pre-constructed image feature database. Then, the reports corresponding to these images are inputted into a report generator of the pre-trained model, obtaining the distributions of retrieved reports. RIN generates final reports by integrating prediction distributions of the pre-trained model and the average distributions of retrieved reports, thereby enhancing the accuracy and reliability of the generated report. Comprehensive experimental results demonstrate that RIN significantly enhances clinical efficacy in chest X-rays report generation task. Compared to the current state-of-the-art methods, it achieves competitive results.

## 1 INTRODUCTION

Information technology has made significant contributions to modern medicine. Non-invasive medical imaging technologies, such as X-rays, ultrasound and MRI, have become essential tools for disease diagnosis and patient monitoring (Panayides et al., 2020). These imaging techniques provide high-resolution images of internal structures, helping in the early detection and diagnosis of various conditions. Since medical images usually involve multiple anatomical structures and pathological features, clinical practice requires specialized radiologists to interpret and write reports.

In this context, deep learning technology has made significant progress in automatic medical report generation, particularly in the chest X-rays (Chen et al., 2020; 2022; Liu et al., 2021c) report generation. However, one of the main challenges in this field is achieving cross-modal consistency between medical images and their corresponding reports (Li et al., 2018; Liu et al., 2021b; Li et al., 2020; 2024). Existing methods have demonstrated exemplary performance in generating reports of high textual quality, but it is often difficult to fully capture the semantic information in medical images (Kaur & Mittal, 2022; Park et al., 2020; Pellegrini et al., 2023; Divya et al., 2024). Specifically, medical images are highly similar, with essential areas taking up only a more minor part, while medical reports' textual structure and content are highly repetitive. This situation leads to the generated medical report that achieves high textual similarity with reference reports but ignores the accurate description of disease diagnosis. Such accuracy in disease diagnosis is crucial. In the medical field, insufficient diagnostic accuracy can have severe consequences (Kalra, 2004; Fabri & Zayas-Castro, 2008; Sarker & Vincent, 2005). For example, missed diagnoses of lung cancer are relatively common, and such oversights can lead to delays in disease assessment and the initiation of treatment (Turkington et al., 2002). In order to capture the semantic information in medical images, several initial approaches have been explored, including the use of contrastive information (Liu et al., 2021d; Li et al., 2023) to focus on the abnormal regions, construct knowledge graphs to provide additional supervision signals (Zhang et al., 2020; Huang et al., 2023), introduce detectors to direct

identification of medical observations (Pino et al., 2021; Tanida et al., 2023; Li et al., 2024). These methods rely on explicit prior knowledge, such as high-quality annotated data (Pino et al., 2021; Tanida et al., 2023) or professional expertise (Li et al., 2019; Zhang et al., 2023), which is currently lacking in medical report tasks (Liu et al., 2021e; Li et al., 2023). Furthermore, these methods generally inject information by making complex adjustments to the attention modules (Liu et al., 2021e; Li et al., 2023), resulting in a training process that requires high computational overhead. Given these considerations, a crucial question is:

*Can we design a general method to enhance clinical efficacy without explicit prior knowledge and training?*

In this work, we propose a training-free Retrieval Information injectioN (RIN) method that aims to generate accurate and effective reports by simulating the process of Multidisciplinary Consultation. In clinical practice, the Multidisciplinary Consultation by multiple experts' diagnoses and jointly analyzing the patient's condition helps reduce the likelihood of misdiagnosis (Sigl et al., 2023). This approach is widely applied in fields such as radiology and pathology (Kane et al., 2007; Mallory et al., 2015) Inspired by this collaborative approach, we proposed a retrieval method that does not rely on explicit prior knowledge. Specifically, we retrieved images similar to the target image from the database and used the corresponding reports as retrieved-reports for the target image. This approach simulates the process of multiple experts jointly analyzing cases during the expert consultation. Drawing from the experience of contrastive decoding that can inject information without training, we inject the retrieved retrieved-reports information directly into the pre-trained medical report generation model in a training-free manner. The pre-trained model generates reports by integrating its predictions and the retrieved information, thereby enhancing the accuracy and reliability of the final generated report.

In summary, our main contributions are as follows:

 • We proposed a retrieval strategy that simulates the Multidisciplinary Consultation by extracting information from similar cases, thereby enhancing the accuracy of generated reports.

 • We introduce a training-free information injection method that requires only adjusting the report's distribution of the generation stage without additional training.

 • We demonstrated the effectiveness of our method across two distinct medical report generation tasks. The results showed that our method could significantly improve the clinical efficacy of generated reports while not reducing too much textual quality.

## 2 RELATED WORK

### 2.1 MEDICAL REPORT GENERATION

Early work on automatic medical report generation typically employed CNN-RNN structures (Jing et al., 2017; Yin et al., 2019). Recently, transformer models have demonstrated their vast potential in medical diagnostics within multi-modal domains (Xu et al., 2023; Chen et al., 2020; 2022; Alfarghaly et al., 2021). Although these methods have demonstrated exemplary performance in generating reports of high textual quality, they still faced a challenge in the cross-modal consistency between medical images and reports (Li et al., 2018; Liu et al., 2021b; Li et al., 2020; 2024). Specifically, medical images are highly similar, with essential areas taking up only a more minor part, while medical reports' textual structure and content are highly repetitive. Much of the existing work is influenced by previous image caption work. It focuses more on improving textual quality, ignoring the accurate description of critical information such as diseases and equipment within the medical images. However, in medical report generation tasks, textual quality is often unimportant. Tanida et al. (2023) found that using lowercase can significantly enhance the textual quality of radiology report generation. Some recent works have aimed at aligning medical images with reports. These works can be divided into four main categories. The first is using contrastive information (Liu et al., 2021d; Li et al., 2023) to focus on the abnormal regions. This contrast can come from image-image (Liu et al., 2021d) or image-report (Li et al., 2023). Liu et al. (2021d) compares the current input image with normal images to distill the contrastive information. Li et al. (2023) built an Image-Report Contrastive Loss (IRC) to activate radiology reporting by encouraging the positive image-report pairs to have similar representations in contrast to the negative pairs. The second is constructing

knowledge graphs to provide additional supervision signals and incorporating knowledge into the model through cross-attention (Zhang et al., 2020; Huang et al., 2023). Huang et al. (2023) proposed a Knowledge-injected U-Transformer (KiUT) to learn multi-level visual representation and adaptively distill the information with contextual and clinical knowledge for word prediction. The third is introducing detectors to direct identification of medical observations. Such detectors include recognition image classifiers (Pino et al., 2021; Tanida et al., 2023), text classifiers (Liu et al., 2019) , and other detectors (Li et al., 2024) . Li et al. (2024) introduced the concept of counterfactuals, identified key regions by constructing counterfactual images, and effectively fine-tuned the pre-trained LLM through learnable prompts to generate more accurate and comprehensive medical reports. The fourth is retrieval-augmented style of generation(Syeda-Mahmood et al., 2020; Ranjit et al., 2023) . Compared to the previous three works, our method does not rely on proprietary models or explicit prior knowledge but adjusts the distribution by training-free contrast decoding, thereby improving clinical efficacy. Compared with the last work, since we do not rely on fixed templates or classifiers, the generated reports are more natural.

## 2.2 Contrastive Decoding Methods

Contrastive decoding is a training-free method to select the optimal result by evaluating and contrasting outputs from different generation strategies or models. Li et al. (2022) utilized the difference in predicted likelihood between expert and amateur language models (LMs) as a basis for decision-making, constraining the LMs to generate more reliable information. Similar work was used for language detoxification and sentiment-controlled generation (Liu et al., 2021a). Shi et al. (2023) emphasized context information during the generation stage by introducing context-aware decoding. Recent advancements have extended to the visual language models. Zhao et al. (2024) introduced a training-free and API-free framework to guide Large Vision-Language Models (LVLMs) in mitigating hallucinations during the generation process. Wan et al. (2024) employed the mask to generate a comparative image derived from the original image. Contrasting the two different images enhanced the visual prompt. Kornblith et al. (2023) implement classifier-free guidance (Ho & Salimans, 2022) to an auto-regressive captioning model by fine-tuning it to estimate conditional and unconditional caption distributions. Some recent works (Kim et al., 2024; Qiu et al., 2024) have introduced RAG into contrastive decoding methods, aiming to improve the open-domain question answering capabilities of LLM. These existing methods aim to reduce decoding noise in expert models by obtaining contrast coding results between expert models and amateur models, while our approach is to introduce retrieved information as additional knowledge to supplement the results of the expert model.

## 3 Approach

This section introduces the detailed implementation of our proposed training-free Retrieval Information injectioN (RIN) for medical report generation. Figure 1 illustrates that RIN consists of a reports retrieval module, an information injection module and a report filter module.

## 3.1 Reports Retrieval

Our approach is grounded in several critical observations:

• The models often produce nearly identical reports when processing semantically similar samples, leading to information omissions. This phenomenon may stem from the high structural and content similarity among medical reports, which causes the model to cluster similar reports together during training. Models tend to produce averaged outputs across these similar reports, resulting in information loss and inconsistencies. Meanwhile, medical report descriptions are lengthy, and existing methods usually truncate overly long content during the data pre-processing stage, possibly leading to information loss. The diversity of medical report word order exacerbates this problem. Reports containing the same semantics may cause different information omissions when the content is too long due to different word orders.

• The generated reports sometimes focus excessively on localized information within chest X-rays, overlooking other critical medical observations. This issue is particularly pronounced in samples involving external medical devices, where the model tends to provide detailed descriptions of

the device's position and trajectory while neglecting other relevant medical observations. This issue is often data-driven, as certain reports within the dataset concentrate solely on localized information, leading to this bias in the model's outputs.

To address these challenges, we propose an improved strategy: retrieve images similar to the target image from the pre-constructed image feature database and fill in the missing information in the generated report with the reports corresponding to the images. Figure 2 shows the construction of the image feature database and retrieval process.

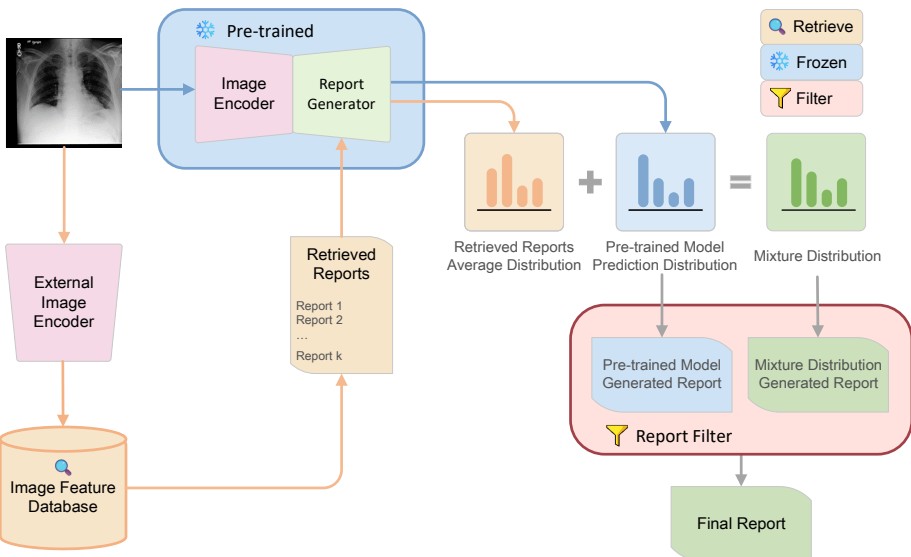

Figure 1: **The workflow of Retrieval Information injectioN (RIN).** RIN consists of a retrieved-reports retrieval module, an information injection module and a report filter module. In step 1, the medical image being processed through a pre-trained Vanilla Model (depicted by the blue line) to generate a predicted report distribution. In step 2, concurrently, the medical image is encoded by an additional external image encoder (depicted by the yellow line) to extract image features. These features retrieve the $k$ most similar images from the image feature database. In step 3, the retrieved-reports corresponding to these similar images are input into the pre-trained report generator, obtaining $k$ retrieved-reports distributions. In step 4, the average of the retrieved-reports is computed and then combined with the predicted report distribution to form a mixture distribution. In step 5, the text decoder independently decodes both predicted report distribution and the mixture distribution into reports. In step 6, the report filter compares the generated reports with the retrieved reports and selects the most similar one as the final report.

**Utilizing Pre-trained Models for Image Encoding** The medical reports are often noisy, and descriptions with different sentences may represent the same content, which increases the difficulty of processing and understanding the report of the model. This challenge makes it harder to retrieve useful retrieved-reports from medical images. Contrastive learning methods, such as CLIP (Radford et al., 2021), can train on large-scale datasets without explicit labels to align images and text. This approach overcomes the issues of prior works Tanida et al. (2023); Li et al. (2024) requiring annotated data or subject to the classifier category. Specifically, we leveraged the image encoder from the BiomedVLP model (Bannur et al., 2023) as external image encoder to extract image features from the training set and built an image feature retrieval database. BiomedVLP is a pre-trained contrastive learning model specifically on the chest X-rays field. The training set sample images are encoded by the image encoder of BiomedVLP to finally obtain a set of 128-dimensional image features $E \in \mathbb{R}^{\text{batch} \times 128}$.

**Utilizing Similarity Calculation for Retrieving Reports** We begin by encoding each medical image using an external image encoder to extract image features. Next, we perform a nearest neighbor search based on cosine similarity to identify the $k$ most similar images from a pre-constructed image

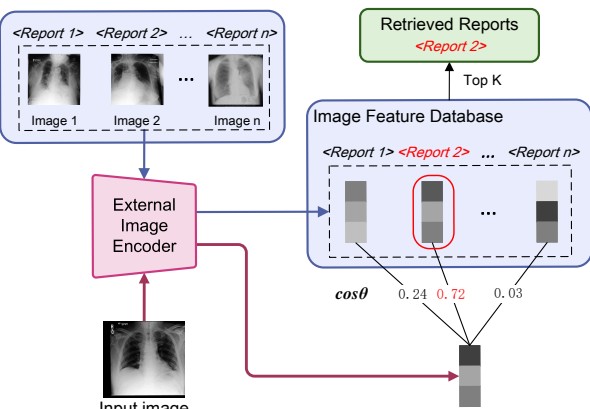

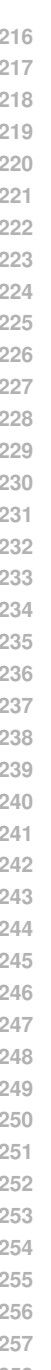

Figure 2: Illustration of the process of retrieved-report retrieval. First, images from the train set are encoded using an external image encoder to generate an image feature database. The target image is then encoded using the same image encoder to obtain its feature representation. Cosine similarity is employed to match the $k$ nearest image features in the database, and the corresponding reports of the matched image features as the retrieved-reports.

feature retrieval database. Finally, the $k$ reports corresponding to these images serve as retrieved-reports to assist in generating the final medical report.

## 3.2 INJECTING RETRIEVAL INFORMATION INTO MEDICAL REPORT GENERATION

For a typical medical report generation problem, given a pre-trained medical report generation model $\theta$, a medical image $I \in \mathbb{R}^{W \times H \times 3}$, and $t$ tokens of report $Y = [y_1, \cdots, y_t]$, this process can be expressed as:

$$y_t \sim p_\theta(y_t \mid I, y_{<t}) \quad \propto \exp\left(\text{logit}_\theta(y_t \mid I, y_{<t})\right) \tag{1}$$

**Training-free Information Injection**  We introduce a training-free approach to inject retrieval information into the pre-trained medical report generation model through adjustments in the decoding process. Firstly, we use PMI (Pointwise Mutual Information) to measure the amount of information sharing between the generated token $Y$ and the retrieval information $C \in \mathbb{R}^{k \times t}$, where $k$ represents $k$ retrieved-reports. We simplify $C \in \mathbb{R}^{k \times t}$ to $C \in \mathbb{R}^t$, given an image $I$ and $t$ tokens of retrieval information $C = [c_1, \cdots, c_t]$ as follows,

$$\text{PMI}(Y; C \mid I) = \log \frac{P(Y, C \mid I)}{P(Y \mid I) \cdot P(C \mid I)} \quad = \log \frac{P(Y \mid I, C)}{P(Y \mid I)} \tag{2}$$

Based on the experience of Classifier-Free Guidance (CFG) (Ho & Salimans, 2022) and Contrastive Region Guidance (CRG) (Wan et al., 2024), the adjustment formula is obtained:

$$Y \sim p_\theta(Y \mid I, C) \quad \propto p_\theta(Y \mid I) \cdot \left(\frac{p_\theta(Y \mid I, C)}{p_\theta(Y \mid I)}\right)^\alpha \tag{3}$$

In practice, for generating a single token $y_t$, we aim to precisely measure the difference between the retrieval information $c_{<t}$ and the previously generated tokens $y_{<t}$. Apply softmax to convert the adjusted logits into probabilities. The following formula is used, where the softmax function is applied to convert the adjusted logits from the first line into probabilities. $p_\theta(y_t \mid I, c_{<t})$ represents the average distribution obtained by averaging over $k$ external retrieved retrieved-reports :

$$y_t \sim p_\theta(y_t \mid I, c_{<t}, y_{<t}) \quad \propto p_\theta(y_t \mid I, y_{<t}) \cdot \left(\frac{p_\theta(y_t \mid I, c_{<t})}{p_\theta(y_t \mid I, y_{<t})}\right)^\alpha \tag{4}$$

$$\sim \text{softmax}\left[(1-\alpha) \cdot \text{logit}_\theta(y_t \mid I, y_{<t}) + \alpha \cdot \text{logit}_\theta(y_t \mid I, c_{<t})\right] \tag{5}$$

$$p_\theta(y_t \mid I, c_{<t}) = \frac{1}{k}\sum_{i=1}^{k} p_\theta(y_t \mid I, c_{i<t}) \tag{6}$$

Here, $\alpha \in [0, 1]$ is a hyperparameter that balances the vanilla model knowledge and the external retrieval knowledge obtained through the retrieval mechanism. A higher value of $\alpha$ indicates stronger control; for example, $\alpha = 1$ represents highly used control, $\alpha = 0$ means standard decoding without control, and $\alpha = \frac{1}{3}$ is suitable for this design, this maintains the same proportions as previous work (Shi et al., 2023; Wan et al., 2024). To observe the influence of different $\alpha$ on information injection, we plot all results from our hyperparameter grid in Figure 5. Besides, we provide a pseudo-code of our information injection in Appendix A.1.1.

## 3.3 REPORT FILTER

Although retrieval information injection (RIN) can effectively enhance the information injection capabilities of generative models, we have observed that RIN may occasionally introduce false positive information that does not exist in the retrieved data. This phenomenon may stem from the characteristics of the auto-regressive generation method. Auto-regressive models generate content sequentially, relying on previously generated outputs, making them prone to propagating errors if any inaccuracies are introduced early in the generation process. To mitigate this issue, we implemented a simple filtering strategy that compares the similarity between the reports generated by the vanilla model, the reports generated after applying RIN, and the retrieved reports to get the report that is most similar to the retrieved information. Specifically, Chexbert(Smit et al., 2020) can automatically encode the radiological report into 14 medical observations. We calculate the average F-1 score of medical observations between the vanilla model generated report, the RIN generated report, and the K retrieved reports using CheXbert. Finally, we select the report with the highest F-1 score from the original or RIN-generated reports as the final output.

## 4 EXPERIMENTS

In this section, we first describe the implementation details. Then, we experimentally validate our method is work on chest X-rays report generation, presenting extensive performance analysis on our retrieved-reports retrieval and information injection modules. Additional details and quantitative findings are in Appendix A.2.

## 4.1 EXPERIMENTAL SETTINGS

We conducted all experiments using one single NVIDIA RTX A5500 GPU.

**Retrieved-reports retrieval module** We utilized the image encoder from the BiomedVLP model pre-trained by Bannur et al. (2023), a contrastive learning model specifically trained on chest X-rays data, to encode images.

**Information injection module** To ensure reproducibility in the contrastive decoding stage, we adopted a greedy decoding strategy and set the beam search width to 4, hyperparameter $\alpha = \frac{1}{3}$, $k = 4$. We employed CvT2DistilGPT2 (Nicolson et al., 2023) as the vanilla model, applying the pre-trained weights from Nicolson et al. (2023).

## 4.2 EVALUATION METRICS

We follow previous work (Liu et al., 2019) in evaluating clinical efficacy (CE). The CE metrics are computed from CheXbert (Smit et al., 2020), a medical report observations classifier that can run on GPUs, providing more accurate and faster extraction of the medical observations compared to CheXpert (Irvin et al., 2019). It can label chest X-rays reports as positive, negative, or uncertain for

each medical observation, then calculates the example-based precision, recall, and F-1 scores of the generated report and corresponding reference report as the CE metrics scores.

At the report level, we follow natural language generation (NLG) metrics, including BLEU (Papineni et al., 2002), METEOR (Banerjee & Lavie, 2005), ROUGE-L (Lin, 2004) and CIDEr (Vedantam et al., 2015). These metrics measure the similarity between generated and reference reports by calculating the overlap of n-grams (i.e., word overlap).

### 4.3 DATASET AND PRE-PROCESSING

**MIMIC-CXR** For the chest X-rays report generation task, we utilized the MIMIC-CXR dataset (Johnson et al., 2019), which was proposed by the Massachusetts Institute of Technology. It is a large-scale de-identified dataset containing 377,110 images and 227,835 radiology reports. The "findings" section of the report includes the observations of radioactive materials. Following previous work (Chen et al., 2020), we excluded samples without the findings section from the dataset, using the findings section as the reference report. The total dataset was adjusted to 276,778 samples. For model training and evaluation, the data were divided into 270,790 training samples, 2,130 validation samples, and 3,858 test samples. To ensure comparability with previous radiology report generation methods, we set the maximum number of words in the report to 60, converted all upper-case letters to lowercase, removed special characters, and replaced words that appeared fewer than three times in the corpus with special unknown tokens. These processing steps are consistent with those used settings of Chen et al. (2020).

### 4.4 MAIN RESULTS

We compare our method with the state-of-the-art report generation systems across automatic chest X-rays report generation. Table 8, Table 2 shows the results. The best ones are marked in bold in the table, and the suboptimal results are marked underlined. We followed the same experimental setup for the automatic chest X-rays report generation task in the original papers, citing their reported results directly.

We compare with the baseline method R2Gen (Chen et al., 2020), CMN (Chen et al., 2022), CA (Liu et al., 2021c), AlignTrans (You et al., 2021), XPRONET(Wang et al., 2022), and the state-of-the-art methods KiUT (Huang et al., 2023), MGSK (Yang et al., 2022), DCL (Li et al., 2023), CvT2DistilGPT2 (as Vanilla model)(Nicolson et al., 2023). As shown in Table 8, our method achieved 0.481, 0.445, and 0.433 in Precision, Recall, and F-1 score, respectively. Compared with the vanilla model, it is improved by **15.1%**, **21.3%**, and **18.0%**, respectively. Although the quality of the NLG metrics has slightly declined, our method still shows strong competitiveness compared with other existing methods. This suggests that our approach has significantly enhanced the clinical efficacy of reports in the chest X-rays automatic report generation task.

| Methods | NLG metrics | | | | | | CE metrics | | |
|---|---|---|---|---|---|---|---|---|---|
| | BLEU-1 | BLEU-2 | BLEU-3 | BLEU-4 | METEOR | ROUGE-L | Precision | Recall | F-1 |
| R2Gen | 0.353 | 0.218 | 0.145 | 0.103 | 0.142 | 0.277 | 0.333 | 0.273 | 0.276 |
| CMN | 0.353 | 0.218 | 0.148 | 0.106 | 0.142 | 0.278 | 0.334 | 0.275 | 0.278 |
| CA | 0.350 | 0.219 | 0.152 | 0.109 | 0.151 | 0.283 | 0.352 | 0.298 | 0.303 |
| AlignTrans | 0.378 | 0.235 | 0.156 | 0.112 | 0.158 | 0.283 | - | - | - |
| XPRONET | 0.344 | 0.215 | 0.146 | 0.105 | 0.138 | 0.279 | - | - | - |
| KiUT | 0.393 | 0.243 | 0.159 | 0.113 | **0.160** | 0.285 | 0.371 | 0.318 | 0.321 |
| MGSK | 0.363 | 0.228 | 0.156 | 0.115 | - | 0.284 | 0.458 | 0.348 | 0.371 |
| DCL | - | - | - | 0.109 | 0.150 | 0.284 | 0.471 | 0.352 | 0.373 |
| CvT2DistilGPT2 | 0.393 | **0.248** | **0.171** | **0.127** | 0.155 | 0.286 | 0.418 | 0.367 | 0.367 |
| +RIN (Ours) | **0.404** | 0.247 | 0.165 | 0.117 | 0.158 | **0.290** | 0.481 | 0.445 | 0.433 |

Table 1: The performance in NLG metrics and CE metrics of our proposed method compared to other competitive methods on the MIMIC-CXR datasets.

Table 2 shows a comparison of our method with the RGRG (Tanida et al., 2023) and CoFE (Li et al., 2024). Both of them only utilized frontal chest X-rays images. Therefore, we extracted frontal images in the test set for a fair comparison. The results indicate that our method demonstrates competitive performance on clinical efficacy metrics compared to state-of-the-art models. Moreover, compared with the vanilla model, our method improves BLEU-1, BLEU-2, METEOR, and ROUGE-

L. It is worth noting that the comparison remains somewhat unfair due to the RGRG splitting the MIMIC-CXR train and test set differently from the previous work.

| Methods | NLG metrics | | | | | | CE metrics | | |
|---|---|---|---|---|---|---|---|---|---|
| | BLEU-1 | BLEU-2 | BLEU-3 | BLEU-4 | METEOR | ROUGE-L | Precision | Recall | F-1 |
| RGRG | 0.373 | **0.249** | **0.175** | **0.126** | 0.168 | 0.264 | 0.461 | 0.475 | 0.447 |
| CoFE | - | - | - | 0.125 | **0.176** | **0.304** | 0.489 | 0.370 | 0.405 |
| CvT2DistilGPT2 | 0.386 | 0.242 | 0.166 | 0.122 | 0.152 | 0.282 | 0.452 | 0.340 | 0.397 |
| +RIN (Ours) | **0.401** | 0.244 | 0.162 | 0.114 | 0.157 | 0.288 | **0.513** | **0.481** | **0.466** |

Table 2: The performance in NLG metrics and CE metrics of our proposed method compared to other competitive methods on MIMIC-CXR datasets' frontal images.

Compared to the vanilla model, our method demonstrates a comprehensive improvement in CE metrics. In terms of NLG metrics, our approach either matches or surpasses the vanilla model in BLEU-1, BLEU-2, METEOR, and ROUGE-L scores, indicating that it generates more lexically precise outputs by selecting words that are closer to the reference text. Additionally, our method shows enhanced performance in capturing overall semantic expression and sentence structure. However, the decrease in BLEU-3 and BLEU-4 scores suggests a limitation in effectively capturing long-range dependencies within our method generated reports.

## 4.5 PERFORMANCE ANALYSIS

**Case Study** To further evaluate the effectiveness of our proposed method, we conducted a comprehensive qualitative analysis comparing the vanilla model with our RIN approach on the MIMIC-CXR dataset. The analysis results show that compared with the vanilla model, our method supplemented the missing information when generating reports and correcting some error information. Specifically, in Figure 3 (a), our approach supplemented crucial details, such as cardiomegaly, pleural effusions and edema, while recorrecting the error information generated by the vanilla model, such as opacity. In Figure 3 (b) shows our method supplemented atelectasis and accurately emphasized the need for further observation of pneumonia. This observation supports the effectiveness of our information retrieval and information injection mechanism.

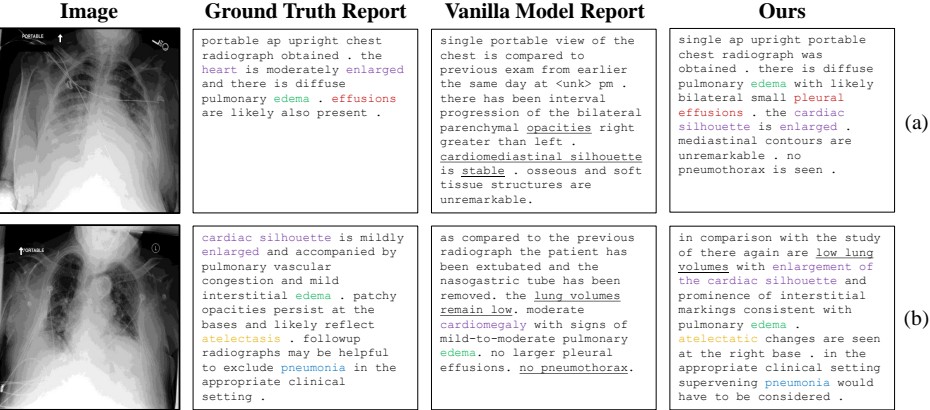

Figure 3: Illustration of reports generated by the vanilla model and our RIN on the MIMIC-CXR dataset. The text in different colors demonstrates the ground truth of medical observations, and the underlining represents the incorrect observation results.

**The Influence of different $k$ values** To further explore the effect of different number of retrieved-report on the clinical efficacy of the generated reports and the text quality. We systematically adjusted the $k$ values ranging from 1 to 10 without using the report filter. Experimental results of Figure 4 reveal the following trends:

• **Variation in CE metrics** We use F1, which combines Recall and Precision to represent the CE metrics. Initially, the F-1 score gradually increases with the increase of the $k$ value, indicating

that increasing the $k$ value within this range can enhance the vanilla model's performance. Lower $k$ values limit the scope of neighboring reports, resulting in more constrained retrieval outcomes. As $k$ values increase, the vanilla model can consider more neighboring reports, capturing more comprehensive information, which helps improve retrieval accuracy. However, when $k$ values continue to increase a certain threshold (in this experiment, $k = 4$), the F-1 score begins to oscillate. This phenomenon suggests that at higher $k$ values, the model starts to incorporate an excessive number of neighboring reports, which may introduce more additional noise or irrelevant information, thus affecting the quality of the retrieval results and causing an oscillation in the F-1 score.

• **Variation in NLG metrics**   Compared to the CE metrics, the NLG metrics show a consistent upward trend as the $k$ value increases, which may be attributed to the fact that as $k$ increases, the retrieved information is more average, making the generated report more semantically, stylistically richer, and more natural.

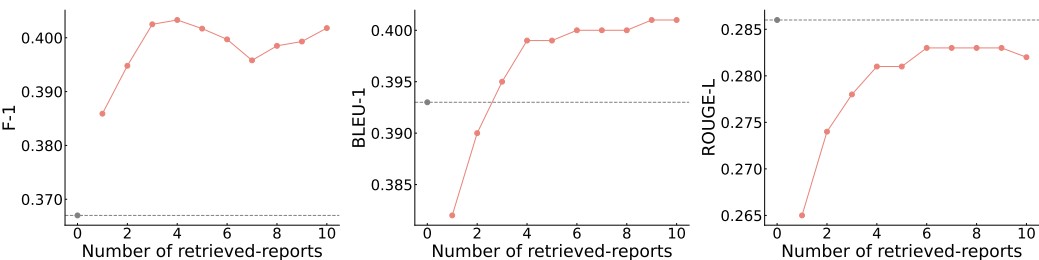

Figure 4: Comparison of metrics over $k$ values.

**The Influence of Different $\alpha$ on Information Injection**   To investigate the influence of hyperparameters on information injection, we further analyze the trade-off associated with the hyperparameter $\alpha$ without using the report filter. In Figure 5, we plot all results from our hyperparameter $\alpha$ grid for $k = 4$. The experiments demonstrate that $\alpha = \frac{1}{3}$ strikes the best balance, maintaining both high text quality and clinical efficacy.

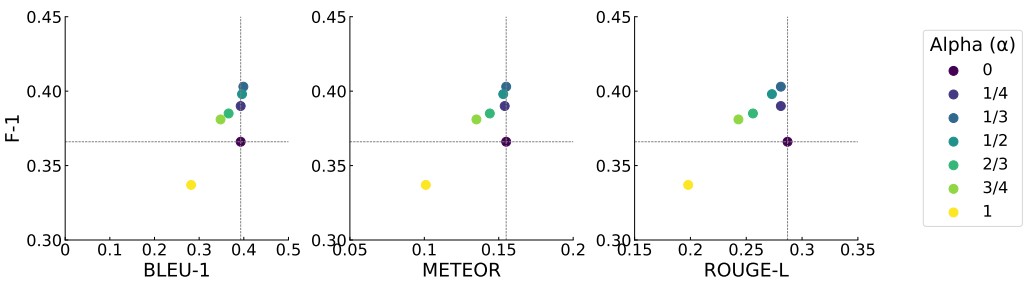

Figure 5: Illustration of all results from our hyperparameter grid.

**Ablation Experiment**   Retrieval information injection can be conceptualized as leveraging the retrieved information as context to enhance the performance of the vanilla model. In order to fully demonstrate the effect of our retrieval information injection, we compared the performance of different modules.

| Pre-trained Model Prediction Distribution | Retrieved-reports Average Distribution | Report Filter | Precision | Recall | F-1 |
|---|---|---|---|---|---|
| ✓ | | | 0.418 | 0.367 | 0.367 |
| | ✓ | | 0.363 | 0.365 | 0.337 |
| ✓ | ✓ | | 0.452 | 0.411 | 0.403 |
| ✓ | ✓ | ✓ | **0.481** | **0.445** | **0.433** |

Table 3: The performance in CE metrics of ablation study on each module.

The "Pre-trained model prediction distribution" refers to the distribution predicted by the vanilla model, "retrieved-reports average distribution" denotes the distribution of retrieved information processed by the vanilla model's report generator Additionally,

the "Report Filter" represents the final report selection strategy mentioned in our methodology. The results are shown in the table 3, Table 4. The observation results show that RIN can effectively enhance the clinical efficacy of the vanilla model while using only retrieval information. Furthermore, the performance is further improved by using the report filter.

| Pre-trained Model Prediction Distribution | Retrieved-reports Average Distribution | Report Filter | BLEU-1 | BLEU-2 | BLEU-3 | BLEU-4 | METEOR | ROUGE-L |
|---|---|---|---|---|---|---|---|---|
| ✓ | | | 0.393 | **0.248** | **0.171** | **0.127** | 0.155 | 0.286 |
| | ✓ | | 0.282 | 0.093 | 0.034 | 0.016 | 0.101 | 0.198 |
| ✓ | ✓ | | 0.400 | 0.245 | 0.162 | 0.114 | 0.157 | 0.288 |
| ✓ | ✓ | ✓ | **0.404** | 0.247 | 0.165 | 0.117 | **0.158** | **0.290** |

Table 4: The performance in NLG metrics of ablation study on each module.

**Algorithm Complexity Analysis**    We analyze the complexity increase introduced by retrieval information injection (RIN) relative to the vanilla model, denoted as $O(n)$. A standard medical report generation model typically consists of an image encoder and a text decoder. Since our method does not involve modifying the image encoder, the complexity of the image encoding stage is consistent with the vanilla model simplified as $O(d)$. We only need to focus on the changes in the complexity of the text generation stage. For the vanilla model, the time complexity of the text decoder can be simplified to $O(t^2 \cdot v)$, where t represents the length of the generated text sequence and v denotes the hidden layer. To introduce our method, during

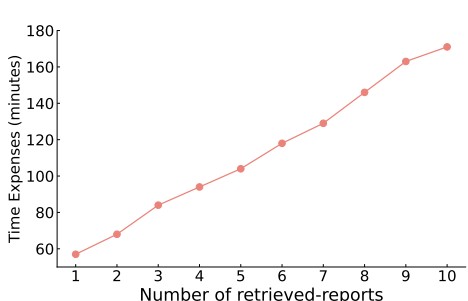

Figure 6: Time expenses of different number of injected retrieved-reports.

the text generation phase, the text decoder's time complexity is adjusted to $O((k+1) \cdot t^2 \cdot v)$, where $k$ represents the number of retrieved retrieved-reports. This adjustment accounts for the additional computation required to calculate the distribution of the retrieved retrieved-reports. As a result, the overall complexity is : $O(n) = O(d) + O((k+1) \cdot t^2 \cdot v)$. Figure 6 shows the change in inference time of RIN when the batch size is 1 and injected the number of retrieved-reports k increases from 1 to 10, further proving that our method only increases the time linearly. Despite the complexity increase, our method provides a training-free injection of retrieval information, enhancing the accuracy and relevance of the generated reports and making this complexity increase reasonable and worthwhile.

## 5    CONCLUSION

In this paper, we introduce a training-free method, Retrieval Information injectioN (RIN), to address the issue of cross-modal consistency between medical images and reports. First, we design a retriever to extract similar images to the target medical image from an image feature database. Then, we employ a contrastive decoding approach, injecting the average distribution of the reports corresponding to the retrieved images as knowledge directly into a pre-trained medical report generation model. Experiments on chest X-rays report generation tasks demonstrate that our approach produces more accurate and clinically efficacy reports.

## 6    LIMITED

The quality of the report generation is affected by the retrieval effect. Poor retrieval performance may not enhance the generation effect of the report generation model and may even have adverse effects. Therefore, in future work, we plan to introduce more accurate retrieval methods to improve the clinical efficacy of generated reports. In addition, the quality of the report in the dataset can also impact the generated reports. Therefore, we aim to refine the contrastive encoding method to better adapt to and handle complex text. With these improvements, we hope to significantly improve the overall quality and accuracy of report generation.

ACKNOWLEDGMENTS

Use unnumbered third level headings for the acknowledgments. All acknowledgments, including those to funding agencies, go at the end of the paper.

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

# A APPENDIX

The structure of our Appendix is as follows. Appendix A.1 provides more details of our RIN framework introduced in section 3 of the main text. Appendix A.2 provides more experimental details and results to help us better understand the capability of RIN. Appendix A.3 analyzes other forms of information injection.

## A.1 IMPLEMENTATION DETAILS OF RIN

In this section, we elaborate on the missing details of RIN in the main text. In particular, we present a summary of the pseudo code for information injection and outline more details of our method's implementation.

### A.1.1 PSEUDO CODE

For clarification, we summarize the pseudo code of Information Injection.

```python
# Input Definitions
# img_embed: Image embedding obtained from an image encoder
# model: Report generation model that predicts token probabilities
# generated_sequence: List to store the sequence of generated tokens,
    initialized as an empty list
# input_ids: List containing the initial input token(s) for generation,
    initialized with [START_TOKEN]
# retrieval_information_ids_list: List of retrieval information tokens
    used to guide the generation process
# alpha: A hyperparameter (0 <= alpha <= 1) that balances the influence
    of the vanilla model and retrieval information
# max_length: The maximum allowable length for the generated sequence
# [END_TOKEN]: A special token that signifies the end of the sequence

# Initialize generated_sequence and decoder_input
generated_sequence = []  # Stores the tokens generated during the process
decoder_input = input_ids.copy()  # Current input to the decoder,
    starting with [START_TOKEN]

# Begin the generation loop
while [END_TOKEN] not in generated_sequence and len(generated_sequence) <
     max_length:
    # Step 1: Predict the next token probabilities using the vanilla
        model
    # The model takes the image embedding (img_embed) and the decoder
        input (decoder_input) as input
    next_token_probabilities = model.predict(img_embed, decoder_input)

    # Step 2: Initialize retrieval information token probabilities to 0
    retrieval_information_next_token_probabilities = 0.0  # This will
        accumulate probabilities guided by retrieval information tokens

    # Step 3: Loop through each retrieval information token set in
        retrieval_information_ids_list
    for retrieval_information_ids in retrieval_information_ids_list:
        # Predict probabilities using the retrieval information token as
            additional input
        # The model predicts how likely the next token is when guided by
            retrieval_information_ids
        retrieval_information_token_probabilities = model.predict(
            img_embed, retrieval_information_ids)

        # Accumulate these probabilities
        retrieval_information_next_token_probabilities +=
            retrieval_information_token_probabilities
```

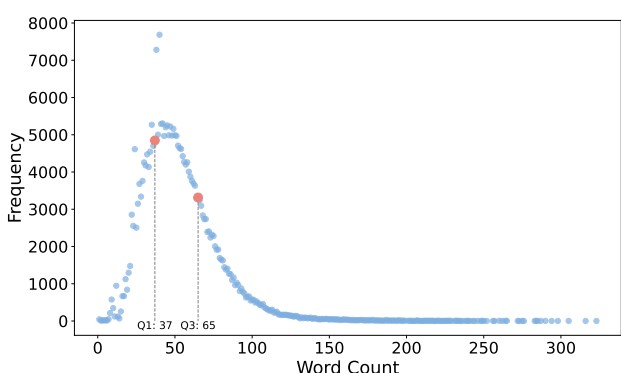

Figure 7: Scatter Plot of Frequency and Word Count.

```
34    # Step 4: Compute the average of retrieval_information token
          probabilities
35    # Normalize the accumulated retrieval information probabilities by
          dividing by the number of retrieval information tokens
36    retrieval_information_next_token_probabilities_average = (
37        retrieval_information_next_token_probabilities / len(
             retrieval_information_ids_list)
38    )
39
40    # Step 5: Combine vanilla model probabilities with retrieval
          information
41    # Use alpha to balance between the two sets of probabilities
42    scores = ((1 - alpha) * next_token_probabilities) + (
43        alpha * retrieval_information_next_token_probabilities_average
44    )
45
46    # Step 6: Select the next token based on the combined scores
47    # The function 'select_token' chooses the next token
48    next_token = select_token(scores)
49
50    # Step 7: Append the selected token to the generated sequence
51    generated_sequence.append(next_token)
52
53    # Step 8: Update the decoder input with the newly selected token
54    decoder_input.append(next_token)
55
56 # Return the final generated sequence
57 return generated_sequence
```

### A.1.2 ADDITIONAL DETAILS OF RIN

**Retrieval dataset settings**

In the MIMIC-CXR dataset, we observed a notable imbalance in the length of reports. To address this issue, we conducted a detailed analysis of the word count for each report in the training set and utilized a scatter plot to visually present the distribution. As shown in Figure 7, the scatter plot analysis revealed that the word counts predominantly fall within a specific range, with the interquartile range (IQR) spanning [37,65]. Within this range, a total of 137,832 samples were identified in the training set. Building on this observation, to enhance retrieval effectiveness, reduce noise interference, and improve retrieval efficiency, we further refined the selection to 71,877 samples falling within the narrower range of [44, 58], thereby constructing a more precise retrieval dataset.

The quality of the retrieved data largely determines the final performance of our approach without using the report filter. Table 5 compares the performance of using all training samples and using only filtered samples as retrieval data in the task of automatically generating chest X-rays reports.

Experimental results show that using filtered samples can significantly improve the effect of report generation, verifying the effectiveness of the report filter in improving the quality of retrieval data.

| | NLG metrics | | | | | | CE metrics | | |
|---|---|---|---|---|---|---|---|---|---|
| Methods | BLEU-1 | BLEU-2 | BLEU-3 | BLEU-4 | METEOR | ROUGE-L | Precision | Recall | F-1 |
| RIN (all training samples) | 0.390 | 0.237 | 0.156 | 0.108 | 0.152 | 0.274 | 0.434 | 0.399 | 0.386 |
| RIN (filtered samples) | 0.400 | 0.245 | 0.162 | 0.114 | 0.157 | 0.288 | 0.452 | 0.411 | 0.403 |

Table 5: The performance in NLG metrics and CE metrics of all training samples and filtered samples on the MIMIC-CXR datasets.

## A.2 MORE RESULTS

In this section, we present additional experimental results to further demonstrate the effectiveness of RIN.

### A.2.1 FURTHER CASE ANALYSIS

In Figure 8, we conducted a further qualitative analysis on the MIMIC-CXR dataset, comparing the vanilla model, retrieved reports, and our approach. The results indicate that compared to the vanilla model, our method effectively supplements the missing information on cardiomegaly and pleural effusion, while accurately describes pleural effusion occurring in bilateral occurrence, and removes the retrieved noise information edema (worsening fluid overload) under our Multidisciplinary Consultation. This observation supports the effectiveness of our information retrieval and information injection mechanisms. However, we also noticed that atelectasis information was commonly found in the retrieved retrieved-reports led to false positive information in the generated reports. Furthermore, since only one retrieved report mentioned opacity, our Multidisciplinary Consultation incorrectly identified this as noise and excluded it, which exposed the limitation of our approach. Therefore, further optimization of the retrieveall mechanism is still necessary to reduce potential false positive results, thereby enhancing the accuracy and reliability of the generated reports.

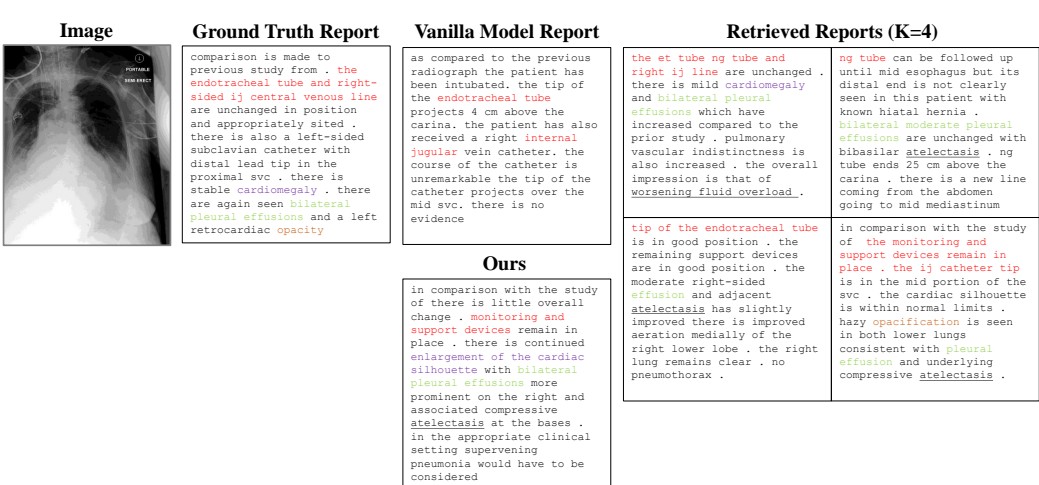

Figure 8: Illustration of the vanilla model, our RIN, and retrieved reports on the MIMIC-CXR dataset. The colored text indicates different medical observations, and underlining indicates false positive information.

### A.2.2 DETAILED CLINICAL EFFICACY METRICS RESULTS

Table 6 detailed results of the clinical efficacy (CE) metrics for each observation as well as micro averaged over all 14 observations.

| Observation | Precision | Recall | F-1 |
|---|---|---|---|
| Micro-Average | 0.522 | 0.485 | 0.503 |
| Atelectasis | 0.388 | 0.402 | 0.395 |
| Cardiomegaly | 0.557 | 0.692 | 0.618 |
| Consolidation | 0.194 | 0.068 | 0.101 |
| Edema | 0.438 | 0.316 | 0.367 |
| Pleural Effusion | 0.620 | 0.633 | 0.626 |
| Enlarged Cardiomediastinum | 0.095 | 0.041 | 0.057 |
| Fracture | 0.059 | 0.020 | 0.030 |
| Lung Lesion | 0.213 | 0.050 | 0.081 |
| Lung Opacity | 0.561 | 0.358 | 0.437 |
| No Finding | 0.222 | 0.467 | 0.301 |
| Pleural Other | 0.148 | 0.033 | 0.054 |
| Pneumonia | 0.189 | 0.121 | 0.148 |
| Pneumothorax | 0.474 | 0.240 | 0.319 |
| Support Devices | 0.745 | 0.804 | 0.773 |

Table 6: The performance of all 14 observations.

### A.3 DIFFERENT STRATEGY OF INFORMATION INJECTION

The construction of retrieval information $C$ directly affects the information injection effect, so we tried different forms of construction and compared them with our method through experiments.

The retrieval information is represented as $C \in \mathbb{R}^{k \times t}$ and the retrieval-reports is represented as $R \in \mathbb{R}^{k \times m}$, where $k$ represents $k$ retrieved-reports. We simplify $C \in \mathbb{R}^{k \times t}$ to $C \in \mathbb{R}^{t}$ and $R \in \mathbb{R}^{k \times m}$ to $R \in \mathbb{R}^{m}$, means $t$ tokens of retrieval information $C = [c_1, \cdots, c_t]$ and $m$ tokens of retrieved-reports $R = [r_1, \cdots, r_m]$

**Form 1**

We directly replace the token generated by the vanilla model before, that is $y_{<t-1}$, with the report token as the injection information. We use padding and truncation to complete or truncate the tokens in $R$ that are less than or more than $t$. At this time $C = [r_1, \cdots, r_{t-1}]$

**Form2**

We inject the complete retrieval information in a prompt-like form. Specifically, when injecting information, we concatenate the retrieved information $R$ with $y_{<t}$ into $C = [r_1, \cdots, r_m, y_1, \cdots, y_{t-1}]$ to generate the next token.

**RIN**

We inject the retrieval information token by token, only replace $y_{=t-1}$, with the report token as the injection information. We use padding and truncation to complete or truncate the tokens in $R$ that are less than or more than $t$. At this time $C = [y_1, \cdots, y_{t-2}, r_{t-1}]$

| Methods | NLG metrics | | | | | | CE metrics | | |
|---|---|---|---|---|---|---|---|---|---|
| | BLEU-1 | BLEU-2 | BLEU-3 | BLEU-4 | METEOR | ROUGE-L | Precision | Recall | F-1 |
| Form1 | 0.390 | 0.241 | 0.160 | 0.111 | 0.152 | 0.285 | 0.447 | 0.393 | 0.391 |
| Form2 | 0.028 | 0.017 | 0.012 | 0.009 | 0.057 | 0.151 | 0.235 | 0.178 | 0.192 |
| RIN | 0.400 | 0.245 | 0.162 | 0.114 | 0.157 | 0.288 | 0.452 | 0.411 | 0.403 |

Table 7: The performance in different information injection.

We compared different information injection methods without using report filter. Table 7 shows the experimental results indicate that our method effectively injects information, whereas Form1 and Form2 fail to achieve similar success. The failure of Form1 may be attributed to its reliance solely on retrieved reports for information, which leads to a loss of memory regarding previously generated tokens by the model. In contrast, the failure of Form2 could stem from the model not being trained to incorporate prompts as input information, resulting in an inability to decode in conjunction with the prompts.

## A.4 REBUTTAL

### A.4.1 FOR REVIEWER FSKN

Thanks to the precious suggestions made by the Reviewer Fskn. These suggestions provide us with a lot of insights and help us improve the quality of our work. We are also highly grateful to the reviewer for dedicating her/his time and effort to help us improve the quality of our paper.

Q1: *Although the improvement in results is significant, there is a lack of intuitive explanation or insight into the source of this improvement.*

A1: Thanks for your comment. As mentioned in the abstract (line 017-019), "The essence of this method lies in fully utilizing similar reports of target images to enhance the performance of pre-trained medical report generation models."

Q2: *Additionally, it remains unclear whether this decode strategy can be applied to other report generation methods.*

A2: In our original manuscripts, we integrated our RINmodule in CvT2DistilGPT2. CvT2DistilGPT2 uses GPT2 as Report Generator. In A4, we integrated our modules to the latest SOTA model PromptMRG(Jin et al., 2024). PromptMRG uses Bert as Report Generator, proving that our method is **Model-Agnostic** and generally applicable to various autoregressive generation methods.

Q3: *If space permits, I suggest moving the details of the INFORMATION INJECTION (currently at the end of the supplementary materials) into the Methods section. Additionally, the current pseudocode is not detailed enough and should be elaborated further.*

A3: Thanks for your suggestion, we will move it to the Methods section later. Besides, We have rewritten the pseudocode, please refer to Appendix A.1.1 in our manuscript.

Q4: *The experimental results in Table 1 do not reach the current state-of-the-art (SOTA) level. The authors could try to combine more advanced methods to verify the stability of the proposed decoding strategy.*

A4:Thank you for providing us with the latest SOTA baseline PromptMRG(Jin et al., 2024). We have supplemented the results of adding our method to the pre-trained PromptMRG model. The detailed parameters are as follows: k=3, $\alpha = 1/3$ (this is the default setting in our paper), beam search within to 3 (this is the default setting in the author's paper(Jin et al., 2024)), and the results are shown in the following table. The experimental results show that our method has achieved 2.8%, 4.1%, and 3.8% improvements in the three CE metrics of precision recall F1, respectively.

| Methods | NLG metrics | | | | | | CE metrics | | |
|---|---|---|---|---|---|---|---|---|---|
| | BLEU-1 | BLEU-2 | BLEU-3 | BLEU-4 | METEOR | ROUGE-L | Precision | Recall | F-1 |
| R2Gen | 0.353 | 0.218 | 0.145 | 0.103 | 0.142 | 0.277 | 0.333 | 0.273 | 0.276 |
| CMN | 0.353 | 0.218 | 0.148 | 0.106 | 0.142 | 0.278 | 0.334 | 0.275 | 0.278 |
| CA | 0.350 | 0.219 | 0.152 | 0.109 | 0.151 | 0.283 | 0.352 | 0.298 | 0.303 |
| AlignTrans | 0.378 | 0.235 | 0.156 | 0.112 | 0.158 | 0.283 | - | - | - |
| XPRONET | 0.344 | 0.215 | 0.146 | 0.105 | 0.138 | 0.279 | - | - | - |
| KiUT | 0.393 | 0.243 | 0.159 | 0.113 | **0.160** | 0.285 | 0.371 | 0.318 | 0.321 |
| MGSK | 0.363 | 0.228 | 0.156 | 0.115 | - | 0.284 | 0.458 | 0.348 | 0.371 |
| DCL | - | - | - | 0.109 | 0.150 | 0.284 | 0.471 | 0.352 | 0.373 |
| CvT2DistilGPT2 | 0.393 | **0.248** | **0.171** | **0.127** | 0.155 | 0.286 | 0.418 | 0.367 | 0.367 |
| +RIN (Ours) | **0.404** | 0.247 | 0.165 | 0.117 | 0.158 | **0.290** | 0.481 | 0.445 | 0.433 |
| PromptMRG* | 0.387 | 0.230 | 0.147 | 0.100 | 0.148 | 0.261 | 0.505 | 0.461 | 0.452 |
| +RIN (Ours) | 0.370 | 0.220 | 0.140 | 0.094 | 0.154 | 0.264 | **0.519** | **0.480** | **0.469** |

Table 8: The performance in NLG metrics and CE metrics of our proposed method compared to other competitive methods on the MIMIC-CXR datasets.

*Since we do not have access to the MIMIC-CXR Database preprocessed by R2Gen, our experiments are conducted directly on the original MIMIC-CXR Database provided by physionet, which results in lower baseline results than the performance in the author's paper.

Q5: *In Table 3, it appears that the proposed retrieved-reports average distribution ... could further strengthen this method.*

A5: Thank you for your suggestion. We will consider introducing a more suitable denoising module in the next version.

Q6: *Does this decoding strategy heavily rely on retrieval accuracy?*

A6: Thanks for your comment. Our decoding strategy depends on, but is not entirely dependent on, retrieval accuracy. RIN generates reports based on the hyperparameter $\alpha$-balanced retrieval information and vanilla model prediction results, We tried experimenting with different external image encoders and distance metrics, and the results showed that even using a simple clip as an external encoder for retrieval can improve CE metrics' performance. However, more accurate retrieval information obviously helps generate more effective results.

| Model | L1 Distance | L2 Distance | Cosine Similarity | Precision | Recall | F-1 |
|---|---|---|---|---|---|---|
| | ✓ | | | 0.456 | 0.424 | 0.412 |
| CLIP | | ✓ | | 0.454 | 0.428 | 0.412 |
| | | | ✓ | 0.454 | 0.422 | 0.410 |
| | ✓ | | | 0.475 | 0.438 | 0.427 |
| BiomedVLP | | ✓ | | 0.481 | 0.447 | 0.434 |
| | | | ✓ | 0.481 | 0.445 | 0.433 |

Table 9: The performance in CE metrics of ablation study on each module.

| Model | L1 Distance | L2 Distance | Cosine Similarity | BLEU-1 | BLEU-2 | BLEU-3 | BLEU-4 | METEOR | ROUGE-L |
|---|---|---|---|---|---|---|---|---|---|
| | ✓ | | | 0.403 | 0.245 | 0.164 | 0.117 | 0.157 | 0.288 |
| CLIP | | ✓ | | 0.403 | 0.246 | 0.165 | 0.118 | 0.157 | 0.288 |
| | | | ✓ | 0.402 | 0.245 | 0.163 | 0.116 | 0.157 | 0.288 |
| | ✓ | | | 0.403 | 0.245 | 0.164 | 0.116 | 0.157 | 0.289 |
| BiomedVLP | | ✓ | | 0.404 | 0.247 | 0.165 | 0.117 | 0.158 | 0.290 |
| | | | ✓ | 0.404 | 0.247 | 0.165 | 0.117 | 0.158 | 0.290 |

Table 10: The performance in NLG metrics of ablation study on each module.

### A.4.2 FOR REVIEWER WUQZ

Thanks to the precious suggestions made by the Reviewer WUqZ. These suggestions provide us with a lot of insights and help us improve the quality of our work. We are also highly grateful to the reviewer for dedicating her/his time and effort to help us improve the quality of our paper.

Q1: *In Section REPORTS RETRIEVAL, ... This work does not further explain the design and effectiveness of the retrieval method. The authors are advised to further validate the effectiveness of the retrieval model.*

A1: Thanks for your comment.To further validate the effectiveness of the retrieval process, we designed an ablation study to compare the performance of different models and distance metrics on the final results. The outcomes are summarized in the table below. The experimental results demonstrate that employing BiomedVLP, a model pretrained on biomedical data, outperforms directly using CLIP for encoding. Additionally, the choice of distance metric has little effect on the results.

| Model | L1 Distance | L2 Distance | Cosine Similarity | Precision | Recall | F-1 |
|---|---|---|---|---|---|---|
| | ✓ | | | 0.456 | 0.424 | 0.412 |
| CLIP | | ✓ | | 0.454 | 0.428 | 0.412 |
| | | | ✓ | 0.454 | 0.422 | 0.410 |
| | ✓ | | | 0.475 | 0.438 | 0.427 |
| BiomedVLP | | ✓ | | 0.481 | 0.447 | 0.434 |
| | | | ✓ | 0.481 | 0.445 | 0.433 |

Table 11: The performance in CE metrics of ablation study on each module.

Q2: *In the ablation study, as shown in Table 4, the model using Pre-trained Model Prediction Distribution, Retrieved-reports Average Distribution, and Report Filter did not achieve the best results in BLEU-2, BLEU-3, and BLEU-4. The authors are advised to further analyze the reasons for the poor performance of the model*

A2: Thanks for your suggestion. When evaluating BLEU scores, it is essential to simultaneously consider additional text metrics such as METEOR and ROUGE. BLEU primarily measures exact n-gram matches, whereas METEOR and ROUGE emphasize semantic relevance and content coverage.

| Model | L1 Distance | L2 Distance | Cosine Similarity | BLEU-1 | BLEU-2 | BLEU-3 | BLEU-4 | METEOR | ROUGE-L |
|---|---|---|---|---|---|---|---|---|---|
| | ✓ | | | 0.403 | 0.245 | 0.164 | 0.117 | 0.157 | 0.288 |
| CLIP | | ✓ | | 0.403 | 0.246 | 0.165 | 0.118 | 0.157 | 0.288 |
| | | | ✓ | 0.402 | 0.245 | 0.163 | 0.116 | 0.157 | 0.288 |
| | ✓ | | | 0.403 | 0.245 | 0.164 | 0.116 | 0.157 | 0.289 |
| BiomedVLP | | ✓ | | 0.404 | 0.247 | 0.165 | 0.117 | 0.158 | 0.290 |
| | | | ✓ | 0.404 | 0.247 | 0.165 | 0.117 | 0.158 | 0.290 |

Table 12: The performance in NLG metrics of ablation study on each module.

As illustrated in Table 4, our approach enhances METEOR and ROUGE scores while exhibiting a decrease in BLEU. This may be because the report generated by our method will cover more semantic information, but the vocabulary in the generated report may have morphological changes.

Moreover, natural language generation (NLG) scores are of limited importance in medical report generation tasks. NLG scores heavily depend on the specific preprocessing applied to reference reports(Tanida et al., 2023). For instance, converting text to lowercase has been shown to substantially improve BLEU scores when compared to uppercase references(Tanida et al., 2023). In contrast, clinical efficacy (CE) metrics are invariant to such preprocessing because they compare the presence or absence of diseases between reference and generated reports(Tanida et al., 2023).

Q3: *The authors are advised to supplement the setting details of hyperparameters, as well as a discussion of model effects using different hyperparame*

A3: Thank you for your suggestion. We have added the hyperparameter $\alpha$ and $k$ introduced in Section 3.2 and Section 4.5 to EXPERIMENTAL SETTINGS. We have discussed the effects of different hyperparameters in Section 4.5 PERFORMANCE ANALYSIS. Please refer to Figure 4 and Figure 5 .

Q4: *Please further explain the differences between the proposed retrieval module and the existing report retrieval methods.*

A4: Thank you for your comment. Existing report retrieval methods can generally be divided into two main categories:

**Methods fully dependent on retrieval**

This approach typically populates a predefined template with the retrieved key information(Syeda-Mahmood et al., 2020). While this ensures consistency, it limits flexibility and adaptability by producing fixed sentence structures. Recent advancements have use of retrieved information as input to large language models (LLMs)(Ranjit et al., 2023) to guide report generation. This enables more natural and diverse outputs but LLMs may struggle to accurately perceive the multiple retrieved reports, leading to biases or omissions(Zhou et al., 2024) in the generated reports.

**Methods integrating retrieval information with report generation models**

These methods incorporate retrieval information into models through mechanisms like attention(Jin et al., 2024). This facilitates more dynamic and context-aware report generation but comes with the drawback of significant training costs.

Our approach generates reports by balancing the knowledge of the vanilla report generation model with the retrieved information in the decoding stage. This allows us to inject additional retrieval information without requiring further training, while preserving the language fluency of the original model.

Q5: *Report generation needs to retrieve k highly relevant reports, how to determine the value of k, and what is the specific value of k used in this paper.*

A5: Thank you for your comment. There are several ways to determine the value of $k$. Here, we introduce two feasible approaches.

Firstly, we need to experiments on the validation set to identify the optimal $k$ for retrieving similar reports. For each validation sample, we retrieved the top-k most similar reports ($k$=1 to 10).

**Evaluate generated reports in validation set to determine the value of $k$**

For different candidate $k$ values, we generate corresponding reports on the validation set and calculate the CE metrics between the generated report and the ground truth report to quantify the generation quality. By comparing the CE performance corresponding to each $k$ value, we finally select the $k$ value with the highest F1 score (best performance) as the determined k value to ensure that the model generation performance is optimal.

**Evaluate retrieved reports to determine the value of $k$**

We calculate the CE metrics between the different retrieved reports of $k$ values and the ground truth reports on the validation set to quantify the generation quality. By comparing the average F1 score performance corresponding to different retrieval reports of $k$ values, we finally select the $k$ value with the highest F1 score (best performance) as the determined k value to ensure that the model generation performance is optimal.

In our manuscript we set $k = 4$.

### A.4.3    FOR REVIEWER KHTY

Thanks to the precious suggestions made by the Reviewer KHTY. These suggestions provide us with a lot of insights and help us improve the quality of our work. We are also highly grateful to the reviewer for dedicating her/his time and effort to help us improve the quality of our paper.

Q1:*There are multiple methods that have taken the RAG ... In general, the relation of retrieval injection to RAG will have to be explained.* A1: Thank you for your suggestion. We have incorporated the mentioned papers into the related work section to ensure a comprehensive contextualization of our study. Below, we provide a detailed explanation of the distinctions between our approach and these referenced methods:

**Report retrieval methods**

• **Methods fully dependent on retrieval** This approach typically populates a predefined template with the retrieved key information(Syeda-Mahmood et al., 2020). While this ensures consistency, it limits flexibility and adaptability by producing fixed sentence structures. Recent advancements have use of retrieved information as input to large language models (LLMs)(Ranjit et al., 2023) to guide report generation. This enables more natural and diverse outputs but LLMs may struggle to accurately perceive the multiple retrieved reports, leading to biases or omissions(Zhou et al., 2024) in the generated reports.

• **Methods integrating retrieval information with report generation models** These methods incorporate retrieval information into models through mechanisms like attention(Jin et al., 2024). This facilitates more dynamic and context-aware report generation but comes with the drawback of significant training costs.

Our approach generates reports by balancing the knowledge of the vanilla report generation model with the retrieved information in the decoding stage. This allows us to inject additional retrieval information without requiring further training, while preserving the language fluency of the original model.

**Contrastive decoding in RAG**

Some recent works(Kim et al., 2024; Qiu et al., 2024) have introduced RAG into contrastive decoding methods, aiming to improve the open-domain question answering capabilities of LLM. This work focuses on mitigating the distractibility issue from both external retrieved documents and parametric knowledge. And these tasks are basically applied to short-form QA tasks. Our job is to generate long reports with clinical efficacy.

Q2: *The terminology used to explain Figure 1 is confusing. You mention text decoders and report generators. Are there referring to the same module or two different modules. If different, this is not reflected in Figure 1.*

A2: Thanks for your comment. They are different. The output of the report generator is a probability distribution, and the text decoder (Beam Search is used in our work) selects the next token based on these probability distributions.

**Q3:** *The use of image encoding features to retrieve similar images needs to be evaluated to see the type of reports retrieved. What is the ratio of overlap of findings of such retrieved reports with the ground truth reports associated with these chest X-rays. Since MIMIC dataset is used, all the chest X-ray images (train-test-validate) should have ground truth reports.*

A3:Thanks for your suggestion. We calculated the clinical efficacy coverage of the retrieval report and the groundtruth of the test set, and the specific results are as follows. We found that the performance of the simple retrieval result is lower than the result of the final generated report, which also reflects that our method is that balances the vanilla model knowledge and the external retrieval knowledge obtained to generate the report.

| Metric | Value |
|---|---|
| Precision | 0.419 |
| Recall | 0.465 |
| F-1 | 0.410 |

Table 13: Performance metrics for CE Precision, Recall, and F1 at the example level.

**Q4:** *Line 296 - Average F-1 score should be based on match to ground truth. Is that what is meant in line 296 or F-1 score is computed relative to which report?*

A4: Thank you for your comment. For each test sample, we calculated the F1 scores between the top-k retrieved reports and the reports generated by the vanilla model as well as the reports generated with RIN. Among vanilla model generated report and RIN generated report, we selected the report with the highest average F1 score as the final report.

**Q5:** *Steps 1-6 described in Figure 1 are not very clear. Is image information used only in step 3 or also in step 5?*

A5: Thanks for your comment. Image information is only used in steps 1 and 2. Step 1 is used as the image input for the vanilla report generation model, and step 2 is used for image feature extraction for the retrieval report.

**Q6:** *Instead of using Bio-VLP for image-to-image matching why not use it directly to retrieve radiology reports as done in earlier papers with CLIP-based retrieval (https://proceedings.mlr.press/v158/endo21a/endo21a.pdf) since Bio-VLP is a multimodal model?*

A6: Thanks for your suggestion, we found that it seems that image-to-text matching is still difficult, which may be due to the diversity of radioactive reports, so we only use the image encoder for image-to-image matching. In order to verify the effectiveness of image-to-image matching. We conducted the following experiments. Experiments show that injecting the image-to-image retrieved reports into the vanilla model can generate higher quality report:

| img2txt | img2img | Precision | Recall | F-1 |
|---|---|---|---|---|
| ✓ | | 0.461 | 0.421 | 0.412 |
| | ✓ | 0.481 | 0.445 | 0.433 |

Table 14: The performance in CE metrics of ablation study on each module.

| img2txt | mg2img | BLEU-1 | BLEU-2 | BLEU-3 | BLEU-4 | METEOR | ROUGE-L |
|---|---|---|---|---|---|---|---|
| ✓ | | 0.397 | 0.240 | 0.159 | 0.112 | 0.154 | 0.285 |
| | ✓ | 0.404 | 0.247 | 0.165 | 0.117 | 0.158 | 0.290 |

Table 15: The performance in NLG metrics of ablation study on each module.

**Q7:** *The use of the term 'distribution' to refer to the generated output from report generator is confusing. Are multiple reports coming out in one step from the report generator?*

A7: Thank you for your comment. The report generator models a probability distribution over the next token, aligning with the interpretation of "distribution" frequently discussed in the provided paper(Qiu et al., 2024). Notably, the generator's ability to produce multiple distinct reports is directly influenced by the configuration of the batch size, which governs the diversity and volume of generated outputs.

**Q8:** *Were the results from CheXBert freshly generated for the datasets by the authors or a reuse of numbers quoted from previous work since the ChexBert using the Allen NLP has some dependencies on older CUDA libraries.*

A8: Thank you for your comment. I apologize if I misunderstood your point. I'll make an effort to understand it better. In our process, we use Chexbert twice: once for filtering and once for evaluating the final effect, and both instances require recalculations.

### A.4.4    FOR REVIEWER GLQW

Thanks to the precious suggestions made by the Reviewer gLqw. These suggestions provide us with a lot of insights and help us improve the quality of our work. We are also highly grateful to the reviewer for dedicating her/his time and effort to help us improve the quality of our paper.

**Q1:** *Has the method been tested on modalities other than chest X-rays, such as MRIs or CT scans, to assess its adaptability and effectiveness?*

A1: Thank you for your suggestion. We attempted to evaluate the effectiveness of our method on the Caption Prediction Task of the ImageCLEFmedical Caption 2023 challenge. The evaluation was conducted on the ROCO V2 dataset, which includes various types of medical images such as ultrasound, X-ray, PET, CT, MRI, and angiography. We incorporated our method into the pretrained MedICap model and present the results. To build the image retrieval database, we used BioMedCLIP instead of BiomedVLP, this is a contrastive learning model pretrained on various medical image types. During the decoding phase, our settings were as follows: k=7, $\alpha = 1/3$ (the default settings in our paper), and beam search within 4 (as reported by the authors). The results are shown in the table below.

We found that existing methods seem to be unable to effectively measure the subtle differences in the generated reports, which may be because these methods were not developed for medical text evaluation. In the absence of methods to evaluate clinical efficacy in the task, we employ the MED-CON metric (Yim et al., 2023) to assess the alignment between generated and referenced reports. MEDCON metric is currently widely used in different types of medical text evaluation(Yim et al., 2024; Van Veen et al., 2023). Different terminological systems may employ varying names or codes to represent the same concept. Within the Unified Medical Language System (UMLS)(Bodenreider, 2004), each medical concept is assigned a distinct Concept Unique Identifier (CUI). MEDCON extracts each medical concept's unique identifier (CUI) in the surgical report through the QuickUMLS (Soldaini & Goharian, 2016) and computes the F1-score to determine the similarity between the UMLS concept sets in predicted and referenced reports. Experiments show that our method can effectively improve the accuracy of medical concept description

| Team Name | Run ID | BERTScore | ROUGE | BLEURT | BLEU | METEOR | CIDEr | CLIPScore |
|---|---|---|---|---|---|---|---|---|
| SSNSheerinKavitha | 4 | 0.544 | 0.087 | 0.215 | 0.075 | 0.026 | 0.014 | 0.687 |
| IUST NLPLAB | 6 | 0.567 | **0.290** | 0.223 | **0.268** | **0.100** | 0.177 | 0.807 |
| Bluefield-2023 | 3 | 0.578 | 0.153 | 0.272 | 0.154 | 0.060 | 0.101 | 0.784 |
| Clef-CSE-GAN-Team | 2 | 0.582 | 0.218 | 0.269 | 0.145 | 0.070 | 0.174 | 0.789 |
| CS Morgan | 10 | 0.582 | 0.156 | 0.224 | 0.057 | 0.044 | 0.084 | 0.759 |
| DLNU CCSE | 1 | 0.601 | 0.203 | 0.263 | 0.106 | 0.056 | 0.133 | 0.773 |
| SSN MLRG | 1 | 0.602 | 0.211 | 0.277 | 0.142 | 0.062 | 0.128 | 0.776 |
| KDE-Lab Med | 3 | 0.615 | 0.222 | 0.301 | 0.156 | 0.072 | 0.182 | 0.806 |
| VCMI | 5 | 0.615 | 0.218 | 0.308 | 0.165 | 0.073 | 0.172 | 0.808 |
| PCLmed | 5 | 0.615 | 0.253 | 0.317 | 0.217 | 0.092 | 0.232 | 0.802 |
| AUEB-NLP-Group | 2 | 0.617 | 0.213 | 0.295 | 0.169 | 0.072 | 0.147 | 0.804 |
| closeAI2023 | 7 | 0.628 | 0.240 | **0.321** | 0.185 | 0.087 | **0.238** | 0.807 |
| CSIRO (MedICap)* | 4 | 0.644 | 0.248 | 0.314 | 0.175 | 0.096 | 0.208 | **0.820** |
| +RIN | / | **0.647** | 0.248 | 0.314 | 0.175 | 0.096 | 0.209 | **0.820** |

Table 16: Performance metrics for different teams (reversed order).

| Methods | Medcon |
|---|---|
| CSIRO (MedICap)* | 0.202 |
| +RIN | 0.245 |

Table 17: Performance metrics for different teams (reversed order).

Q2: *Since medical images are highly similar as mentioned in the paper, is it possible for the workflow to retrieve images that are similar but have distinct symptoms, leading to inaccurate diagnosis?*

A2: Thanks for your comment. The retrieved reports may include false-positive observations, which we address by employing an averaging mechanism during the decoding and report filtering stages. This approach mimics the voting process in expert consensus, aiming to mitigate the impact of such false positives. However, in extreme cases—when the majority of the retrieved reports contain the same false-positive observations—this mechanism may fail. For instance, as illustrated in Appendix A.2, most retrieved reports incorrectly identified false-positive observations of atelectasis, leading RIN to erroneous inclusion of atelectasis in the generated results.

