# OpenReview forum: "Retrieval Information Injection for Enhanced Medical Report Generation"
_ICLR.cc/2025/Conference — Submitted to ICLR 2025_

### Official Review · Reviewer_gLqw · 2024-11-02

**Soundness:** 2
**Presentation:** 3
**Contribution:** 2
**Rating:** 6
**Confidence:** 3

**Summary:**

This paper presents a novel method called Retrieval Information injection (RIN) aimed at enhancing the accuracy and efficacy of automatically generated medical reports. RIN involves retrieving similar images from a pre-constructed database and integrating corresponding report information with the predictions of a pre-trained model. By simulating the process of Multidisciplinary Consultation, the proposed method explores the issues related to cross-modal consistency between medical images and reports, showing competitive clinical efficacy and textual quality (evaluated via CE metrics and NLG metrics).

**Strengths:**

The paper is well-written and insightful. Extensive results demonstrate that RIN improves clinical efficacy in generating reports for chest X-rays, which provides a practical solution to the diagnostic bottleneck caused by physician shortages.

**Weaknesses:**

Despite the paper's clarity, several imprecise arguments and overstatements necessitate revision and clarification:
* The effectiveness of the retrieval process depends heavily on the quality and diversity of the pre-constructed image feature database, which might limit the method's applicability in poorly-documented medical fields.
* The implementation of a report filter in the workflow does not significantly improve the situation where false positive information still exists.
* The paper does not extensively explore the impact of errors in the retrieval process itself as a comparison.

**Questions:**

* Has the method been tested on modalities other than chest X-rays, such as MRIs or CT scans, to assess its adaptability and effectiveness?
* Since medical images are highly similar as mentioned in the paper, is it possible for the workflow to retrieve images that are similar but have distinct symptoms, leading to inaccurate diagnosis?

**Details Of Ethics Concerns:**

Please provide details of your concerns.

---

> ### Author Response · Authors · 2024-11-25
>
> Thanks to the precious suggestions made by the Reviewer gLqw. These suggestions provide us with a lot of insights and help us improve the quality of our work. We are also highly grateful to the reviewer for dedicating her/his time and effort to help us improve the quality of our paper.
>
> Q1: **Has the method been tested on modalities other than chest X-rays, such as MRIs or CT scans, to assess its adaptability and effectiveness?**
>
> A1: Thank you for your suggestion. We attempted to evaluate the effectiveness of our method on the Caption Prediction Task of the ImageCLEFmedical Caption 2023 challenge[1]. The evaluation was conducted on the ROCO V2 dataset[2], which includes various types of medical images such as ultrasound, X-ray, PET, CT, MRI, and angiography. We incorporated our method into the pretrained MedICap model and present the results. To build the image retrieval database, we used BioMedCLIP instead of BiomedVLP, this is a contrastive learning model pretrained on various medical image types. During the decoding phase, our settings were as follows: $k=7$, $\alpha=1/3$ (the default settings in our paper), and beam search within to 4 (as reported by the authors). The results are shown in the table below.
>
>
> | **Team Name**          | **Run ID** | **BERTScore** | **ROUGE** | **BLEURT** | **BLEU** | **METEOR** | **CIDEr** | **CLIPScore** |
> |-------------------------|------------|---------------|-----------|------------|-----------|------------|------------|---------------|
> | SSNSheerinKavitha      | 4          | 0.544         | 0.087     | 0.215      | 0.075     | 0.026      | 0.014      | 0.687         |
> | IUST NLPLAB            | 6          | 0.567         | **0.290** | 0.223      | **0.268** | **0.100**  | 0.177      | 0.807         |
> | Bluefield-2023         | 3          | 0.578         | 0.153     | 0.272      | 0.154     | 0.060      | 0.101      | 0.784         |
> | Clef-CSE-GAN-Team      | 2          | 0.582         | 0.218     | 0.269      | 0.145     | 0.070      | 0.174      | 0.789         |
> | CS Morgan              | 10         | 0.582         | 0.156     | 0.224      | 0.057     | 0.044      | 0.084      | 0.759         |
> | DLNU CCSE              | 1          | 0.601         | 0.203     | 0.263      | 0.106     | 0.056      | 0.133      | 0.773         |
> | SSN MLRG               | 1          | 0.602         | 0.211     | 0.277      | 0.142     | 0.062      | 0.128      | 0.776         |
> | KDE-Lab Med            | 3          | 0.615         | 0.222     | 0.301      | 0.156     | 0.072      | 0.182      | 0.806         |
> | VCMI                   | 5          | 0.615         | 0.218     | 0.308      | 0.165     | 0.073      | 0.172      | 0.808         |
> | PCLmed                 | 5          | 0.615         | _0.253_   | 0.317      | _0.217_   | 0.092      | _0.232_    | 0.802         |
> | AUEB-NLP-Group         | 2          | 0.617         | 0.213     | 0.295      | 0.169     | 0.072      | 0.147      | 0.804         |
> | closeAI2023            | 7          | 0.628         | 0.240     | **0.321**  | 0.185     | 0.087      | **0.238**  | 0.807         |
> | CSIRO (MedICap)*    | 4          | _0.644_       | 0.248     | 0.314      | 0.175     | _0.096_    | 0.208      | **0.820**     |
> | +RIN               | /          | **0.647**     | 0.248     | 0.314      | 0.175     | _0.096_    | 0.209      | **0.820**     |
>
>
> ---
> \*The results except CSIRO (MedICap) and RIN are from ImageCLEFmedical Caption[1]. CSIRO (MedICap) is reproduced by us and the result is slightly higher than the result reported in ImageCLEFmedical Caption[1].
>
> We found that existing methods seem to be unable to effectively measure the subtle differences in the generated reports, which may be because these methods were not developed for medical text evaluation. In the absence of methods to evaluate clinical efficacy in the task, we employ the MEDCON metric[3] to assess the alignment between generated and referenced reports. MEDCON metric is currently widely used in different types of medical text evaluation[4,5].  Different terminological systems may employ varying names or codes to represent the same concept. Within the Unified Medical Language System (UMLS), each medical concept is assigned a distinct Concept Unique Identifier (CUI). MEDCON extracts each medical concept's unique identifier (CUI) in the surgical report through the QuickUMLS[6] and computes the F1-score to determine the similarity between the UMLS concept sets in predicted and referenced reports. Experiments show that our method can effectively improve the accuracy of medical concept description.
>
>
> | **Methods**            | **Medcon** |
> |-------------------------|------------|
> | CSIRO (MedICap)*        | 0.202      |
> | +RIN                   | 0.245      |

---

> ### Author Response · Authors · 2024-11-25
>
> Q2: **Since medical images are highly similar as mentioned in the paper, is it possible for the workflow to retrieve images that are similar but have distinct symptoms, leading to inaccurate diagnosis?**
>
> A2: Thanks for your comment. The retrieved reports may include false-positive observations, which we address by employing an averaging mechanism during the decoding and report filtering stages. This approach mimics the voting process in expert consensus, aiming to mitigate the impact of such false positives. However, in extreme cases—when the majority of the retrieved reports contain the same false-positive observations—this mechanism may fail. For instance, as illustrated in **Appendix A.2.1**, most retrieved reports incorrectly identified false-positive observations of atelectasis, leading RIN to erroneous inclusion of atelectasis in the generated results.
>
> [1]https://www.imageclef.org/2023/medical/caption
>
> [2]Rückert, Johannes, et al. "Rocov2: Radiology objects in context version 2, an updated multimodal image dataset." Scientific Data 11.1 (2024): 688.
>
> [3]Yim, Wen-wai, et al. "Aci-bench: a novel ambient clinical intelligence dataset for benchmarking automatic visit note generation." Scientific Data 10.1 (2023): 586.
>
> [4]Van Veen, Dave, et al. "Clinical text summarization: adapting large language models can outperform human experts." Research Square (2023).
>
> [5]Yim, Wen-wai, et al. "Dermavqa: A multilingual visual question answering dataset for dermatology." International Conference on Medical Image Computing and Computer-Assisted Intervention. Cham: Springer Nature Switzerland, 2024.
>
> [6]Soldaini, Luca, and Nazli Goharian. "Quickumls: a fast, unsupervised approach for medical concept extraction." MedIR workshop, sigir. 2016.

---

> > ### Comment · Reviewer_gLqw · 2024-11-27
> > **Thank you for the responses**
> >
> > Thank you for your detailed response and for conducting evaluations on diverse modalities using the ROCO V2 dataset. The authors have addressed my concerns. However, as authors noted, this mechanism may fail in cases where false-positive observations dominate the retrieved reports. I will keep my original rating unchanged.

---

> ### Author Response · Authors · 2024-11-27
>
> We sincerely appreciate the reviewer’s thoughtful time and effort in thoroughly evaluating our paper and offering invaluable feedback.

---

### Official Review · Reviewer_KHTY · 2024-11-03

**Soundness:** 2
**Presentation:** 1
**Contribution:** 2
**Rating:** 3
**Confidence:** 5

**Summary:**

The paper presents a retrieval-augmented style of generation (RAG) of radiology reports for chest X-rays. Specifically, radiology reports from similar chest X-rays are used to refine the output of a report generative model using contrastive decoding RAG approaches. A selection is then made between the refined textual output,and  the original predicted report without RAG from image directly by comparing to the retrieved reports from similar chest X-rays.

The base report generator used is CV2DistilGPT2, the chest x-image image similarity is computed using Bio-VLP encoding, and contrastive decoding is implemented using greedy beam search of width 4.

Overall, the paper offers limited novelty with a rehash of many of the ideas used in other contemporary papers. Added to this, the use of 14 finding label based training limits the practical utility of such methods also seen from the low F-1 scores even for these 14 findings.

**Strengths:**

The basic idea that RAG improves report generation has been shown for the chest X-ray report case.
Extensive experiments (take up nearly 50% of the paper) have been performed analyzing factors they deemed important to test. Lexical, semantic and clinical efficacy scores were used for evaluation.

**Weaknesses:**

There are multiple methods that have taken the RAG (retrieval augmented generation) approaches to report generation that haven't been mentioned in related work nor compared to them. These include methods those that first classify the findings in chest X-rays and retrieve related reports that match in terms of findings which are more likely to be better in the selection of reports than those based purely on image features[1]-[2]. At least a comparison to these is validated. Similarly, two other papers referenced below use RAG for chest X-ray report generation which will need comparison[2]-[3]. Finally, the differences with other contrastive decoding approached to RAG may have to be discussed [3]-[4]. In general, the relation of retrieval injection to RAG will have to be explained.

[1]https://arxiv.org/abs/2007.13831
[2] https://arxiv.org/abs/2305.03660 - uses RAG
[3] https://arxiv.org/html/2408.01084v1 - Adaptive contrastive decoding in RAG
[4] https://arxiv.org/html/2406.17519v1 - Entropy based contrastive ensemble

**Questions:**

1. The terminology used to explain Figure 1 is confusing. You mention text decoders and report generators. Are there referring to the same module or two different modules. If different, this is not reflected in Figure 1.

2. The use of image encoding features to retrieve similar images needs to be evaluated to see the type of reports retrieved. What is the ratio of overlap of findings of such retrieved reports with the ground truth reports associated with these chest X-rays. Since MIMIC dataset is used, all the chest X-ray images (train-test-validate) should have ground truth reports.

3. Line 296 - Average F-1 score should be based on match to ground truth. Is that what is meant in line 296 or F-1 score is computed relative to which report?

4. Steps 1-6 described in Figure 1 are not very clear. Is image information used only in step 3 or also in step 5?

5. Instead of using Bio-VLP for image-to-image matching why not use it directly to retrieve radiology reports as done in earlier papers with CLIP-based retrieval (https://proceedings.mlr.press/v158/endo21a/endo21a.pdf) since Bio-VLP is a multimodal model?

6. The use of the term 'distribution' to refer to the generated output from report generator is confusing. Are multiple reports coming out in one step from the report generator?

7. Were the results from CheXBert freshly generated for the datasets by the authors or a reuse of numbers quoted from previous work since the ChexBert using the Allen NLP has some dependencies on older CUDA libraries.

**Details Of Ethics Concerns:**

None.

---

> ### Author Response · Authors · 2024-11-25
>
> Thanks to the precious suggestions made by the Reviewer KHTY. These suggestions provide us with a lot of insights and help us improve the quality of our work. We are also highly grateful to the reviewer for dedicating her/his time and effort to help us improve the quality of our paper.
>
> Q1:**There are multiple methods that have taken the RAG... In general, the relation of retrieval injection to RAG will have to be explained.**
>
> A1: Thank you for your suggestion. We have incorporated the mentioned papers into the related work section to ensure a comprehensive contextualization of our study. Below, we provide a detailed explanation of the distinctions between our approach and these referenced methods:
>
> **Report retrieval methods**
>
> Existing report retrieval methods can generally be divided into two main categories:
>
> 1. Methods fully dependent on retrieval. This approach typically populates a predefined template with the retrieved key information[2]. While this ensures consistency, it limits flexibility and adaptability by producing fixed sentence structures. Recent advancements have introduced the use of retrieved information as input to large language models (LLMs)[4] to guide report generation. This enables more natural and diverse outputs but LLMs may struggle to accurately perceive the multiple retrieved reports, leading to biases or omissions[6] in the generated reports.
>
> 2. Methods integrating retrieval information with report generation models. These methods incorporate retrieval information into models through mechanisms like attention[2]. This facilitates more dynamic and context-aware report generation but comes with the drawback of significant training costs.
>
> Compared to the first method, **our method effectively ensures linguistic fluency** in report generation. This is achieved by balancing the inherent knowledge of the vanilla report generation model with the retrieved information. In contrast to the second method, **our method is training-free**. By leveraging the contrastive decoding strategy that adjusts the probability distribution during the decoding stage, our method eliminates the need for additional training. As a result, our method addresses the challenges faced by the other two approaches, namely the inability to generate coherent reports or the need for additional training costs.
>
> **Contrastive decoding in RAG**
>
> Some recent works[4,5] have introduced RAG into contrastive decoding methods, aiming to improve the open-domain question answering capabilities of LLM. These work focuses on mitigating the distractibility issue from both external retrieved documents and parametric knowledge. And these tasks are basically applied to short-form QA tasks. Our job is to generate long reports with clinical efficacy.
>
> Q2: **The terminology used to explain Figure 1 is confusing. You mention text decoders and report generators. Are there referring to the same module or two different modules? If different, this is not reflected in Figure 1.**
>
> A2: Thanks for your comment. They are different. The output of the report generator is a probability distribution, and the text decoder (Beam Search is used in our work) selects the next token based on these probability distributions.
>
> Q3: **The use of image encoding features to retrieve similar images needs to be evaluated to see the type of reports retrieved. What is the ratio of overlap of findings of such retrieved reports with the ground truth reports associated with these chest X-rays. Since MIMIC dataset is used, all the chest X-ray images (train-test-validate) should have ground truth reports.**
>
> A3: Thanks for your suggestion. We calculated the clinical efficacy coverage of the retrieval report and the groundtruth of the test set, and the specific results are as follows. We found that the performance of the simple retrieval result is lower than the result of the final generated report, which also reflects that our method is that balances the vanilla model knowledge and the external retrieval knowledge obtained to generate the report.
>
> | Metric                   | Precision | Recall | F-1 |
> |--------------------------|-------------------------|----------------------|-----------------|
> | **Value**                | 0.419   | 0.465  | 0.410 |

---

> ### Author Response · Authors · 2024-11-25
>
> Q4:**Line 296 - Average F-1 score should be based on match to ground truth. Is that what is meant in line 296 or F-1 score is computed relative to which report?**
>
> A4: Thank you for your comment. For each test sample, we calculated the F1 scores between the top-k retrieved reports and the reports generated by the vanilla model as well as the reports generated with RIN. Among vanilla model generated report and RIN generated report, we selected the report with the highest average F1 score as the final report.
>
> Q5: **Steps 1-6 described in Figure 1 are not very clear. Is image information used only in step 3 or also in step 5?**
>
> A5: Thanks for your comment. Image information is only used in steps 1 and 2. Step 1 is used as the image input for the vanilla report generation model, and step 2 is used for image feature extraction for the retrieval report.
>
> Q6:**Instead of using Bio-VLP for image-to-image matching why not use it directly to retrieve radiology reports as done in earlier papers with CLIP-based retrieval (https://proceedings.mlr.press/v158/endo21a/endo21a.pdf) since Bio-VLP is a multimodal model?**
>
> A6: Thanks for your suggestion, we found that it seems that image-to-text matching is still difficult, which may be due to the diversity of radioactive reports, so we only use the image encoder for image-to-image matching. In order to verify the effectiveness of image-to-image matching. We conducted the following experiments in the same settings: $k=4$, $\alpha=1/3$, and beam search within to 4. Experiments show that injecting the image-to-image retrieved reports into the vanilla model can generate higher quality report:
>
> | img2txt | img2img | BLEU-1 | BLEU-2 | BLEU-3 | BLEU-4 | METEOR | ROUGE-L | Precision | Recall | F-1   |
> |---------|---------|--------|--------|--------|--------|--------|---------|-----------|--------|-------|
> | ✔       |         | 0.397  | 0.240  | 0.159  | 0.112  | 0.154  | 0.285   | 0.461     | 0.421  | 0.412 |
> |         | ✔       | 0.404  | 0.247  | 0.165  | 0.117  | 0.158  | 0.290   | 0.481     | 0.445  | 0.433 |
>
>
> Q7: **The use of the term 'distribution' to refer to the generated output from report generator is confusing. Are multiple reports coming out in one step from the report generator?**
>
> A7: Thank you for your comment. The report generator models a probability distribution over the next token, aligning with the interpretation of "distribution" frequently discussed in the provided paper [5]. Notably, the generator's ability to produce multiple distinct reports is directly influenced by the configuration of the batch size, which governs the diversity and volume of generated outputs.
>
> Q8: **Were the results from CheXBert freshly generated for the datasets by the authors or a reuse of numbers quoted from previous work since the ChexBert using the Allen NLP has some dependencies on older CUDA libraries.**
>
> A8: Thank you for your comment. I apologize if I misunderstood your point. I’ll make an effort to understand it better. In our process, we use Chexbert twice: once for filtering and once for evaluating the final effect, and both instances require recalculations.
>
> [1]Syeda-Mahmood, Tanveer, et al. "Chest x-ray report generation through fine-grained label learning." Medical Image Computing and Computer Assisted Intervention–MICCAI 2020: 23rd International Conference, Lima, Peru, October 4–8, 2020, Proceedings, Part II 23. Springer International Publishing, 2020.
>
> [2]Jin, Haibo, et al. "Promptmrg: Diagnosis-driven prompts for medical report generation." Proceedings of the AAAI Conference on Artificial Intelligence. Vol. 38. No. 3. 2024.
>
> [3]Ranjit, Mercy, et al. "Retrieval augmented chest x-ray report generation using openai gpt models." Machine Learning for Healthcare Conference. PMLR, 2023.
>
> [4]Kim, Youna, et al. "Adaptive Contrastive Decoding in Retrieval-Augmented Generation for Handling Noisy Contexts." arXiv preprint arXiv:2408.01084 (2024).
>
> [5]Qiu, Zexuan, et al. "Entropy-based decoding for retrieval-augmented large language models." arXiv preprint arXiv:2406.17519 (2024).
>
> [6]Zhou, Yujia, et al. "Trustworthiness in retrieval-augmented generation systems: A survey." arXiv preprint arXiv:2409.10102 (2024).

---

> > ### Author Response · Authors · 2024-11-29
> >
> > Dear Reviewer KHTY:
> >
> > Thanks a lot for your efforts in reviewing this paper. We tried our best to address the mentioned concerns. Are there unclear explanations or remaining problems? We will try our best to address them.
> >
> > Kind regards,
> >
> > Authors.

---

> > > ### Author Response · Authors · 2024-12-01
> > > **Further Discussion**
> > >
> > > Dear Reviewer KHTY:
> > >
> > > Thanks a lot for your efforts in reviewing this paper. We tried our best to address the mentioned concerns. Are there unclear explanations or remaining problems? We will try our best to address them.
> > >
> > > Kind regards,
> > >
> > > Authors.

---

### Official Review · Reviewer_WUqZ · 2024-11-03

**Soundness:** 2
**Presentation:** 2
**Contribution:** 2
**Rating:** 3
**Confidence:** 4

**Summary:**

This work discusses the challenge of automatically generating medical reports to alleviate the diagnostic bottleneck caused by physician shortage. Existing methods excel at producing high-textual-quality reports. However, the high similarity among medical images and the homogeneity of medical reports' structure and content hinder the full capture of semantic information in images. To overcome this, the authors propose a training-free Retrieval Information injection (RIN) method, which enhances pre-trained medical report generation models by utilizing similar reports of target images. This approach aims to improve the accuracy and clinical efficacy of the generated reports without requiring additional training.

**Strengths:**

1. This work complements the results of the expert model by introducing retrieval information as additional knowledge. Unlike previous generation methods that rely only on image or text information, this method incorporates information from retrieved similar reports and injects the retrieved information into a pre-trained medical report generation model by adjusting the decoding process. This way of combining retrieval and generation is relatively novel in the field of medical report generation.

2. The structure of the manuscript is clear, logical and coherent, and the connection between the contents of each part is natural. When introducing the method, the authors elaborate on the key steps of retrieval information acquisition, injection, and adjustment of decoding process, so that the reader can clearly understand the working principle of the method.

**Weaknesses:**

1. In Section REPORTS RETRIEVAL, the authors claim that medical reports are often noisy, which increases the difficulty of processing and understanding model reports, making it more difficult to retrieve useful retrieval reports from medical images. This work infuses the retrieved information into the pre-trained medical report generation model, which means that the quality of report generation is affected by the retrieval effect. If the retrieved information is inaccurate or not highly relevant, then the injected retrieved information may not be effective in improving the quality of report generation, and may even have a negative impact. This work does not further explain the design and effectiveness of the retrieval method. The authors are advised to further validate the effectiveness of the retrieval model.

2. In the ablation study, as shown in Table 4, the model using Pre-trained Model Prediction Distribution, Retrieved-reports Average Distribution, and Report Filter did not achieve the best results in BLEU-2, BLEU-3, and BLEU-4. The authors are advised to further analyze the reasons for the poor performance of the model

3. The authors are advised to supplement the setting details of hyperparameters, as well as a discussion of model effects using different hyperparameters.

**Questions:**

1. Please further explain the differences between the proposed retrieval module and the existing report retrieval methods.
2. Report generation needs to retrieve k highly relevant reports, how to determine the value of k, and what is the specific value of k used in this paper.

---

> ### Author Response · Authors · 2024-11-25
>
> Thanks to the precious suggestions made by the Reviewer WUqZ. These suggestions provide us with a lot of insights and help us improve the quality of our work. We are also highly grateful to the reviewer for dedicating her/his time and effort to help us improve the quality of our paper.
>
> Q1: **In Section REPORTS RETRIEVAL, ... This work does not further explain the design and effectiveness of the retrieval method. The authors are advised to further validate the effectiveness of the retrieval model.**
>
> A1: Thanks for your comment. To further validate the effectiveness of the retrieval process, we designed an ablation study to compare the performance of different models and distance metrics on the final results. We conducted the following experiments in the same settings: $k=4$, $\alpha=1/3$, and beam search within to 4. The outcomes are summarized in the table below. The experimental results demonstrate that employing BiomedVLP, a model pretrained on biomedical data, outperforms directly using CLIP for encoding. Additionally, the choice of distance metric has little effect on the results.
>
> | Model      | L1 Distance | L2 Distance | Cosine Similarity | BLEU-1 | BLEU-2 | BLEU-3 | BLEU-4 | METEOR | ROUGE-L | Precision | Recall | F-1   |
> |------------|-------------|-------------|-------------------|--------|--------|--------|--------|--------|---------|-----------|--------|-------|
> |            | ✓           |             |                   | 0.403  | 0.245  | 0.164  | 0.117  | 0.157  | 0.288   | 0.456     | 0.424  | 0.412 |
> | CLIP       |             | ✓           |                   | 0.403  | 0.246  | 0.165  | 0.118  | 0.157  | 0.288   | 0.454     | 0.428  | 0.412 |
> |            |             |             | ✓                 | 0.402  | 0.245  | 0.163  | 0.116  | 0.157  | 0.288   | 0.454     | 0.422  | 0.410 |
> ||
> |            | ✓           |             |                   | 0.403  | 0.245  | 0.164  | 0.116  | 0.157  | 0.289   | 0.475     | 0.438  | 0.427 |
> | BiomedVLP  |             | ✓           |                   | 0.404  | 0.247  | 0.165  | 0.117  | 0.158  | 0.290   | 0.481     | 0.447  | 0.434 |
> |            |             |             | ✓                 | 0.404  | 0.247  | 0.165  | 0.117  | 0.158  | 0.290   | 0.481     | 0.445  | 0.433 |
>
>
>
> Q2: **In the ablation study, as shown in Table 4, the model using Pre-trained Model Prediction Distribution, Retrieved-reports Average Distribution, and Report Filter did not achieve the best results in BLEU-2, BLEU-3, and BLEU-4. The authors are advised to further analyze the reasons for the poor performance of the model**
>
> A2: Thanks for your suggestion. When evaluating BLEU scores, it is essential to simultaneously consider additional text metrics such as METEOR and ROUGE. BLEU primarily measures exact n-gram matches, whereas METEOR and ROUGE emphasize semantic relevance and content coverage. As illustrated in Table 4, our approach enhances METEOR and ROUGE scores while exhibiting a decrease in BLEU. This may be because the report generated by our method will cover more semantic information, but the vocabulary in the generated report may have morphological changes.
>
> Moreover, natural language generation (NLG) scores are of limited importance in medical report generation tasks. NLG scores heavily depend on the specific preprocessing applied to reference reports[1]. For instance, converting text to lowercase has been shown to substantially improve BLEU scores when compared to uppercase references[1].  In contrast, clinical efficacy (CE) metrics are invariant to such preprocessing because they compare the presence or absence of diseases between reference and generated reports[1].
>
> Q3: **The authors are advised to supplement the setting details of hyperparameters, as well as a discussion of model effects using different hyperparame**
>
> A3: Thank you for your suggestion. We would like to kindly point out that the relevant hyperparameters, $\alpha$ and $k$, were introduced in Sections 3.2 and 4.5 of the **original manuscript**. To enhance clarity, we have now included these hyperparameters in the EXPERIMENTAL SETTINGS section. Additionally, we have analyzed the effects of different hyperparameters in Section 4.5 PERFORMANCE ANALYSIS of the **original manuscript**. For further details, please refer to Figure 4 and Figure 5.

---

> ### Author Response · Authors · 2024-11-25
>
> Q4: **Please further explain the differences between the proposed retrieval module and the existing report retrieval methods.**
>
> A4: Thank you for your comment. Existing report retrieval methods can generally be divided into two main categories:
>
> **Methods fully dependent on retrieval.** This approach typically populates a predefined template with the retrieved key information[2]. While this ensures consistency, it limits flexibility and adaptability by producing fixed sentence structures. Recent advancements have use of retrieved information as input to large language models (LLMs)[4] to guide report generation. This enables more natural and diverse outputs but LLMs may struggle to accurately perceive the multiple retrieved reports, leading to biases or omissions [5] in the generated reports.
>
> **Methods integrating retrieval information with report generation models.** These methods incorporate retrieval information into models through mechanisms like attention[2]. This facilitates more dynamic and context-aware report generation but comes with the drawback of significant training costs.
>
> Compared to the first method, **our method effectively ensures linguistic fluency** in report generation. This is achieved by balancing the inherent knowledge of the vanilla report generation model with the retrieved information. In contrast to the second method, **our method is training-free**. By leveraging the contrastive decoding strategy that adjusts the probability distribution during the decoding stage, our method eliminates the need for additional training. As a result, our method addresses the challenges faced by the other two approaches, namely the inability to generate coherent reports or the need for additional training costs.
>
> Q5: **Report generation needs to retrieve $k$ highly relevant reports, how to determine the value of $k$, and what is the specific value of $k$ used in this paper.**
>
> A5: Thanks for your comment. We will address your two questions in order. The first question pertains to the issue of determining the value of $k$, and the second question relates to the parameters of specific value of $k$ used in our manuscript.
>
> There are several ways to determine the value of $k$. Here, we introduce two feasible approaches.
>
> We need to experiments on the validation set to identify the optimal $k$ for retrieving similar reports. For each validation sample, we retrieved the top-k most similar reports ($k$=1 to 10).
>
> **Evaluate generated reports in validation set to determine the value of $k$.** For different candidate $k$ values, we generate corresponding reports on the validation set and calculate the CE metrics between the generated report and the ground truth report to quantify the generation quality. By comparing the CE performance corresponding to each $k$ value, we finally select the $k$ value with the highest F1 score (best performance) as the determined k value to ensure that the model generation performance is optimal.
>
> **Evaluate retrieved reports to determine the value of $k$.** We calculate the CE metrics between the different retrieved reports of $k$ values and the ground truth reports on the validation set to quantify the generation quality. By comparing the average F1 score performance corresponding to different retrieval reports of $k$ values, we finally select the $k$ value with the highest F1 score (best performance) as the determined k value to ensure that the model generation performance is optimal.
>
> We next address the concern about the **specific value of $k$** used in this manuscript. We kindly remind the reviewer, in fact, we have introduced the relevant parameter of the $k$ in the original manuscript. In our manuscript we set $k=4$.
>
> [1]Tanida, Tim, et al. "Interactive and explainable region-guided radiology report generation." Proceedings of the IEEE/CVF Conference on Computer Vision and Pattern Recognition. 2023.
>
> [2]Syeda-Mahmood, Tanveer, et al. "Chest x-ray report generation through fine-grained label learning." Medical Image Computing and Computer Assisted Intervention–MICCAI 2020: 23rd International Conference, Lima, Peru, October 4–8, 2020, Proceedings, Part II 23. Springer International Publishing, 2020.
>
> [3]Jin, Haibo, et al. "Promptmrg: Diagnosis-driven prompts for medical report generation." Proceedings of the AAAI Conference on Artificial Intelligence. Vol. 38. No. 3. 2024.
>
> [4]Ranjit, Mercy, et al. "Retrieval augmented chest x-ray report generation using openai gpt models." Machine Learning for Healthcare Conference. PMLR, 2023.
>
> [5]Zhou, Yujia, et al. "Trustworthiness in retrieval-augmented generation systems: A survey." arXiv preprint arXiv:2409.10102 (2024).

---

> > ### Author Response · Authors · 2024-11-29
> >
> > Dear Reviewer WUqZ:
> >
> > Thanks a lot for your efforts in reviewing this paper. We tried our best to address the mentioned concerns. Are there unclear explanations or remaining problems? We will try our best to address them.
> >
> > Kind regards,
> >
> > Authors.

---

> > > ### Author Response · Authors · 2024-12-01
> > > **Further Discussion**
> > >
> > > Dear Reviewer WUqZ:
> > >
> > > Thanks a lot for your efforts in reviewing this paper. We tried our best to address the mentioned concerns. Are there unclear explanations or remaining problems? We will try our best to address them.
> > >
> > > Kind regards,
> > >
> > > Authors.

---

### Official Review · Reviewer_Fskn · 2024-11-05

**Soundness:** 3
**Presentation:** 2
**Contribution:** 2
**Rating:** 5
**Confidence:** 4

**Summary:**

In this work, the authors introduce a training-free Retrieval Information injectioN (RIN) method to enhance the accuracy and effectiveness of medical report generation. Inspired by the Multidisciplinary Consultation process, where multiple experts collaboratively diagnose and analyze a patient's condition, the method retrieves similar images from a database and uses their corresponding reports as references. This method proposes an innovative decoding strategy that enhances the quality of the decoded text by injecting retrieved similar reports into the decoding process. This method has been validated through extensive experiments, demonstrating the effectiveness of both the model and the proposed specific strategies.

**Strengths:**

1.The proposed method is simple and straightforward, distinct from traditional contrastive decoding strategies, and its effectiveness has been demonstrated. The fact that it requires no training makes it particularly attractive.
2.The experiments are comprehensive. In addition to the state-of-the-art (SOTA) analysis, the authors conducted extensive analytical experiments, including ablation studies and evaluations of various parameters within the method.

**Weaknesses:**

1.Although the improvement in results is significant, there is a lack of intuitive explanation or insight into the source of this improvement. Additionally, it remains unclear whether this decode strategy can be applied to other report generation methods.
2.If space permits, I suggest moving the details of the INFORMATION INJECTION (currently at the end of the supplementary materials) into the Methods section. Additionally, the current pseudocode is not detailed enough and should be elaborated further.
3.The experimental results in Table 1 do not reach the current state-of-the-art (SOTA) level [1]. The authors could try to combine more advanced methods to verify the stability of the proposed decoding strategy.
4.In Table 3, it appears that the proposed retrieved-reports average distribution may introduce noise during the decoding process (as discussed in section 3.3), also evidenced by comparing it with the results in the full version (the last row), which included the report filter. Although it achieved significant improvement compared to the baseline (the first row), addressing such adverse effects in certain situations could further strengthen this method.
5.Does this decoding strategy heavily rely on retrieval accuracy?


Reference:
[1] Jin H, Che H, Lin Y, et al. Promptmrg: Diagnosis-driven prompts for medical report generation[C]//Proceedings of the AAAI Conference on Artificial Intelligence. 2024, 38(3): 2607-2615.

**Questions:**

refer to the weaknesses.

---

> ### Author Response · Authors · 2024-11-25
>
> Thanks to the precious suggestions made by the Reviewer Fskn. These suggestions provide us with a lot of insights and help us improve the quality of our work. We are also highly grateful to the reviewer for dedicating her/his time and effort to help us improve the quality of our paper.
>
> Q1: **Although the improvement in results is significant, there is a lack of intuitive explanation or insight into the source of this improvement.**
>
> A1: Thanks for your comment. As mentioned in the **abstract** (line 017-019), "The essence of this method lies in fully utilizing similar reports of target images to enhance the performance of pre-trained medical report generation models."
>
> Q2: **Additionally, it remains unclear whether this decode strategy can be applied to other report generation methods.**
>
> A2: In our original manuscript, we integrated our RINmodule in CvT2DistilGPT2. CvT2DistilGPT2 uses GPT2 as Report
> Generator. In **A4**, we integrated our modules to the latest SOTA model PromptMRG[1]. PromptMRG uses Bert as Report
> Generator, proving that our method is **Model-Agnostic** and generally applicable to various autoregressive generation methods.
>
> Q3: **If space permits, I suggest moving the details of the INFORMATION INJECTION (currently at the end of the supplementary materials) into the Methods section. Additionally, the current pseudocode is not detailed enough and should be elaborated further.**
>
> A3: Thanks for your suggestion, we will move it to the Methods section later. Besides, We have rewritten the pseudocode, please refer to  Appendix A1.1 in our manuscript.
>
> Q4: **The experimental results in Table 1 do not reach the current state-of-the-art (SOTA) level. The authors could try to combine more advanced methods to verify the stability of the proposed decoding strategy.**
>
> A4: Thank you for providing us with the latest SOTA baseline PromptMRG[1]. We have supplemented the results of adding our method to the pre-trained PromptMRG model. The detailed parameters are as follows: $k$=3, $\alpha=1/3$ (this is the default setting in our paper), beam search within to 3 (this is the default setting in the author's paper[1]), and the results are shown in the following table. The experimental results show that our method has achieved **2.8\%**, **4.1\%** and **3.8\%** improvements in the three CE metrics of precision, recall and F1, respectively.
>
>
> | Methods           | BLEU-1  | BLEU-2  | BLEU-3  | BLEU-4  | METEOR        | ROUGE-L       | Precision | Recall | F-1   |
> |--------------------|---------|---------|---------|---------|---------------|---------------|-----------|--------|-------|
> | R2Gen             | 0.353   | 0.218   | 0.145   | 0.103   | 0.142         | 0.277         | 0.333     | 0.273  | 0.276 |
> | CMN               | 0.353   | 0.218   | 0.148   | 0.106   | 0.142         | 0.278         | 0.334     | 0.275  | 0.278 |
> | CA                | 0.350   | 0.219   | 0.152   | 0.109   | 0.151         | 0.283         | 0.352     | 0.298  | 0.303 |
> | AlignTrans        | 0.378   | 0.235   | 0.156   | 0.112   | _0.158_       | 0.283         | -         | -      | -     |
> | XPRONET           | 0.344   | 0.215   | 0.146   | 0.105   | 0.138         | 0.279         | -         | -      | -     |
> | KiUT              | _0.393_ | 0.243   | 0.159   | 0.113   | **0.160**     | 0.285         | 0.371     | 0.318  | 0.321 |
> | MGSK              | 0.363   | 0.228   | 0.156   | 0.115   | -             | 0.284         | 0.458     | 0.348  | 0.371 |
> | DCL               | -       | -       | -       | 0.109   | 0.150         | 0.284         | 0.471     | 0.352  | 0.373 |
> | CvT2DistilGPT2    | _0.393_ | **0.248** | **0.171** | **0.127** | 0.155     | _0.286_       | 0.418     | 0.367  | 0.367 |
> | +RIN (Ours)       | **0.404** | _0.247_ | _0.165_ | _0.117_ | _0.158_      | **0.290**     | 0.481     | 0.445  | 0.433 |
> | PromptMRG*        | 0.387   | 0.230   | 0.147   | 0.100   | 0.148         | 0.261         | _0.505_   | _0.461_| _0.452_|
> | +RIN (Ours)       | 0.370   | 0.220   | 0.140   | 0.094   | 0.154         | 0.264         | **0.519** | **0.480** | **0.469** |
>
>
> \*Since we do not have access to the MIMIC-CXR Database preprocessed by R2Gen, our experiments are conducted directly on the original MIMIC-CXR Database provided by physionet, which results in lower baseline results than the performance in the author's paper.

---

> ### Author Response · Authors · 2024-11-25
>
> Q5: **In Table 3, it appears that the proposed retrieved-reports average distribution...could further strengthen this method.**
>
> A5: Thank you for your suggestion. We will consider introducing a more suitable denoising module in the next version.
>
> Q6: **Does this decoding strategy heavily rely on retrieval accuracy?**
>
> A6: Thanks for your comment. Our decoding strategy depends on, but is not entirely dependent on, retrieval accuracy. RIN generates reports based on the hyperparameter $\alpha$-balanced retrieval information and vanilla model prediction results. We tried experimenting with different external image encoders and distance metrics, and the results showed that even using a simple clip as an external encoder for retrieval can improve CE metrics’ performance. We conducted the following experiments in the same settings: $k=4$, $\alpha=1/3$, and beam search within to 4.  However, more accurate retrieval information obviously helps generate more effective results.
>
>
>
> | Model      | L1 Distance | L2 Distance | Cosine Similarity | BLEU-1 | BLEU-2 | BLEU-3 | BLEU-4 | METEOR | ROUGE-L | Precision | Recall | F-1   |
> |------------|-------------|-------------|-------------------|--------|--------|--------|--------|--------|---------|-----------|--------|-------|
> |            | ✓           |             |                   | 0.403  | 0.245  | 0.164  | 0.117  | 0.157  | 0.288   | 0.456     | 0.424  | 0.412 |
> | CLIP       |             | ✓           |                   | 0.403  | 0.246  | 0.165  | 0.118  | 0.157  | 0.288   | 0.454     | 0.428  | 0.412 |
> |            |             |             | ✓                 | 0.402  | 0.245  | 0.163  | 0.116  | 0.157  | 0.288   | 0.454     | 0.422  | 0.410 |
> ||
> |            | ✓           |             |                   | 0.403  | 0.245  | 0.164  | 0.116  | 0.157  | 0.289   | 0.475     | 0.438  | 0.427 |
> | BiomedVLP  |             | ✓           |                   | 0.404  | 0.247  | 0.165  | 0.117  | 0.158  | 0.290   | 0.481     | 0.447  | 0.434 |
> |            |             |             | ✓                 | 0.404  | 0.247  | 0.165  | 0.117  | 0.158  | 0.290   | 0.481     | 0.445  | 0.433 |
>
>
>
> [1]Jin, Haibo, et al. "Promptmrg: Diagnosis-driven prompts for medical report generation." Proceedings of the AAAI Conference on Artificial Intelligence. Vol. 38. No. 3. 2024.

---

> > ### Author Response · Authors · 2024-11-29
> >
> > Dear Reviewer Fskn:
> >
> > Thanks a lot for your efforts in reviewing this paper. We tried our best to address the mentioned concerns. Are there unclear explanations or remaining problems? We will try our best to address them.
> >
> > Kind regards,
> >
> > Authors.

---

> > > ### Author Response · Authors · 2024-12-01
> > > **Further Discussion**
> > >
> > > Dear Reviewer Fskn:
> > >
> > > Thanks a lot for your efforts in reviewing this paper. We tried our best to address the mentioned concerns. Are there unclear explanations or remaining problems? We will try our best to address them.
> > >
> > > Kind regards,
> > >
> > > Authors.

---

### Comment · Area_Chair_LHsg · 2024-11-29
**Please review the author response**

Dear Reviewers,

If you have not done so, could you review the author response and let them know if you are satisfied with it or if you have any additional questions?

Kind regards,

Your AC

---

### Meta-Review · Area_Chair_LHsg · 2024-12-21

**Metareview:**

This work proposes a method to improve medical report generation. The key idea is to retrieve the reports of the images similar to the input image and inject the information on these reports into the generator to enhance the quality of the output report. Experimental study is conducted to demonstrate the effectiveness of the proposed method. Reviewers comment that this work is straightforward and attractive, has comprehensive experiments, and is well structured and easy to follow. At the same time, reviewers raise issues related to the lack of insight on improvement, the performance with respect to the latest methods, the adverse effect of noise, the reliance on retrieval accuracy, the differences from existing methods, the need of more evaluation on retrieved reports, limited novelty, and so on. The authors provide a rebuttal. The rebuttal does address some of the concerns such as the generality issue, the differences from existing report retrieval methods, experimental settings, and the way of using Bio-VLP. However, the answers to some issues are not informative enough and some issues are not well answered. The key unresolved concerns are 1) the effectiveness and robustness of the retrieval process and how it exactly improves the quality of generated report, 2) the adverse effect of noise and the effectiveness of report filtering shall be better analysed, and 3) the performance on NLG metrics is not well improved or even becomes worse, but this is not sufficiently analysed. By reading the submission, the reviews, the rebuttals, and the message sent by the authors, AC can see that this work has its merits, especially in clearly identifying the issues of current report generation methods and motivating this work. Meanwhile, AC also agrees with the reviewers on the raised concerns. Most of the final ratings are on the negative side. Considering all the factors, this work in its current form cannot be recommended for acceptance. It is hoped that the reviews could help to further improve the quality of this work.

**Additional Comments On Reviewer Discussion:**

Reviewers raise issues related to the lack of insight on improvement, the performance with respect to the latest methods, the adverse effect of noise, the reliance on retrieval accuracy, the differences from existing methods, the need of more evaluation on retrieved reports, limited novelty, and so on. The authors provide a rebuttal. The rebuttal does address some of the concerns such as the generality issue, the differences from existing report retrieval methods, experimental settings, and the way of using Bio-VLP. However, the answers to some issues are not informative enough and some issues are not well answered. The key unresolved concerns are 1) the effectiveness and robustness of the retrieval process and how it exactly improves the quality of generated report, 2) the adverse effect of noise and the effectiveness of reporting filtering shall be better analysed, and 3) the performance on NLG metrics is not well improved or even becomes worse, but this is not sufficiently analysed. By reading the submission, the reviews, the rebuttals, and the message sent by the authors, AC can see that this work has its merits, especially in clearly identifying the issues of current report generation methods and motivating this work. Meanwhile, AC also agrees with the reviewers on the raised concerns. Most of the final ratings are on the negative side. Considering all the factors, this work in its current form cannot be recommended for acceptance.

---

### Decision · Program_Chairs · 2025-01-22

Reject